# Efficient analysis of mammalian polysomes in cells and tissues using Ribo Mega-SEC

**Harunori Yoshikawa[1†], Mark Larance[1,2†], Dylan J Harney[2], Ramasubramanian Sundaramoorthy[1], Tony Ly[1,3], Tom Owen-Hughes[1], Angus I Lamond[1]***

[1]Centre for Gene Regulation and Expression, School of Life Sciences, University of Dundee, Dundee, United Kingdom; [2]Charles Perkins Centre, School of Life and Environmental Sciences, University of Sydney, Sydney, Australia; [3]Wellcome Centre for Cell Biology, University of Edinburgh, Edinburgh, United Kingdom

**Abstract** We describe Ribo Mega-SEC, a powerful approach for the separation and biochemical analysis of mammalian polysomes and ribosomal subunits using Size Exclusion Chromatography and uHPLC. Using extracts from either cells, or tissues, polysomes can be separated within 15 min from sample injection to fraction collection. Ribo Mega-SEC shows translating ribosomes exist predominantly in polysome complexes in human cell lines and mouse liver tissue. Changes in polysomes are easily quantified between treatments, such as the cellular response to amino acid starvation. Ribo Mega-SEC is shown to provide an efficient, convenient and highly reproducible method for studying functional translation complexes. We show that Ribo Mega-SEC is readily combined with high-throughput MS-based proteomics to characterize proteins associated with polysomes and ribosomal subunits. It also facilitates isolation of complexes for electron microscopy and structural studies.
DOI: https://doi.org/10.7554/eLife.36530.001

**\*For correspondence:**
a.i.lamond@dundee.ac.uk

[†]These authors contributed equally to this work

**Competing interests:** The authors declare that no competing interests exist.

## Introduction

The ribosome is a large RNA-protein complex, comprising four ribosomal RNAs (rRNAs) and >80 ribosomal proteins (RPs), which coordinates mRNA-templated protein synthesis. In human cells, the 80S ribosome has a molecular weight of ~4.3 MDa and a diameter of 250–300 Å (*Khatter et al., 2015*), which is larger than ribosomes in either yeast cells (~3.3 MDa), or in bacteria (~2.3 MDa) (*Melnikov et al., 2012*). During translation, multiple ribosomes can simultaneously engage the same mRNA to form a 'polysome' (*Noll, 2008*; *Warner and Knopf, 2002*; *Warner et al., 1963*). In mammalian cells, it is likely that most protein translation takes place on polysomes, rather than via mRNA translation by a single ribosome.

The biochemical analysis of gene expression and regulation benefits from the ability to isolate and characterize these very large polysome structures. For example, polysome profiling has recently become a popular technique for the analysis of the 'translatome', that is the set of mRNA species being actively translated in polysomes (*King and Gerber, 2016*; *Piccirillo et al., 2014*). Polysome profiling involves the separation and isolation of polysomes away from free ribosomal subunits. This is typically achieved using a sucrose density gradient (SDG) step, prior to mRNA analysis with the enriched polysome fraction.

SDG fractionation, a method introduced in the 1960s (*Britten and Roberts, 1960*; *Warner et al., 1963*), remains the near ubiquitous approach currently used to isolate and analyse polysomes. However, SDG analyses have limitations and can be time-consuming to set up and perform. This includes

the long ultracentrifugation step (typically 3–20 hr) required, which can affect the structure of the ribosome and result in loss of weakly associated proteins (*Simsek et al., 2017*). SDG analysis also requires dedicated hardware for profile generation and fraction collection (*Gandin et al., 2014*). Multiple steps in the SDG workflow can also introduce sources of technical variation (e.g. gradient formation between centrifugation tubes, differences in starting position and collection of fractions etc), leading to an overall decrease in reproducibility and affecting resolution and accuracy (*Ingolia et al., 2009*).

An alternative approach to SDG for isolation of polysomes and free ribosome subunits is to use affinity purification in extracts of cells expressing tagged RPs (*Heiman et al., 2014*; *Inada et al., 2002*; *Simsek et al., 2017*). However, polysomes and ribosomal subunits are co-isolated by this technique and cannot easily be separated. Moreover, the over-expression of RPs either with, or without tags, is known to induce formation of sub-complexes, because of inefficient assembly into ribosomes (*Simsek et al., 2017*). Unassembled ribosomal proteins are rapidly degraded by the ubiquitin-proteasome system (*Lam et al., 2007*; *Sung et al., 2016a*; *Sung et al., 2016b*; *Warner, 1977*).

Arguably, the most efficient and reproducible modern method for fractionation-based biochemical analysis is provided by ultra High-Pressure Liquid Chromatography (uHPLC). This provides exceptional reproducibility, in comparison with other methods, from the use of electronically controlled autosamplers for sample injection and automatic fraction-collection components. Despite these potential advantages, uHPLC methods have not been developed for the effective chromatographic separation of polysomes and ribosome subunits (*Maguire et al., 2008*; *Trauner et al., 2011*). In particular, the very large size of polysomes has been viewed as being outside the effective separation range of size exclusion chromatography methods.

To provide a more efficient and reproducible methodology that can facilitate the isolation of polysomes from mammalian cells and tissues, we have developed an uHPLC Size Exclusion Chromatography (SEC) method for studying very large intracellular structures and protein complexes (Mega-SEC). Here, we demonstrate the application of Mega-SEC to separate translation complexes extracted either from human cell lines, or from mouse liver tissue. We show efficient separation and collection of polysomes, 80S ribosomes, 60S and 40S ribosomal subunits using Ribo Mega-SEC uHPLC runs of ~15 min, with high reproducibility. We also combine Ribo Mega-SEC with downstream electron microscopy and high-throughput proteomic analysis to characterize isolated polysomes.

## Results

To prepare lysates for polysome separations, we employed the standard method of polysome extraction used in traditional SDG analysis. The lysis buffer contained the standard reagents used to stabilize polysomes (cycloheximide, RNase inhibitors and $Mg^{2+}$) (*Klein et al., 2004*; *Ruan et al., 1997*; *Schneider-Poetsch et al., 2010*; *Shi et al., 2017*; *Strezoska et al., 2000*). In addition, the buffer needed to contain a non-denaturing (either non-ionic or zwitterionic) detergent to solubilize the cell membranes. The non-ionic detergent Triton X-100 is the most widely employed in biochemical studies on ribosomes, including SDG analyses (*Ingolia et al., 2012*; *Ingolia et al., 2009*; *Olsnes, 1970*; *Shi et al., 2017*). However, Triton X-100 has a low critical micelle concentration (CMC), and is therefore not ideal for use in an SEC separation buffer. Heparin has been widely used when preparing extracts for polysome analyses but is not always essential and as shown below is best avoided for specific applications, such as MS-based analysis of isolated complexes.

We therefore compared polysome profiles using SDG of lysates prepared in a buffer containing either Triton X-100, or the zwitterionic detergent CHAPS, which is also used for ribosome analysis and whose CMC is sufficiently high such that it can be effectively used in SEC separation buffers below its CMC (*Hjelmeland, 1980*). This showed a similar recovery of polysomes, including halfmers and ribosomal subunits from HeLa cells for both Triton X-100 and CHAPS (*Figure 1—figure supplement 1*). We therefore selected CHAPS for all subsequent experiments, based on its high CMC and because it is known to solubilize protein complexes without denaturation (*Hjelmeland, 1980*). We note that with the HeLa cells and culture conditions used here, which supported rapid cell growth, there was a high polysome:monosome (P/M) ratio, that is with relatively low levels of 80S ribosomes,

as shown both by SDG and SEC analyses. We used these culture conditions in the experiments described below.

To efficiently resolve polysomes and ribosomal subunits from cell extracts by SEC, we next optimized the choice of pore size for the SEC column and the maximum flow rate. First, we compared the SEC chromatograms of HeLa cell lysates separated using three different SEC columns with pore sizes of either 300, 1,000, or 2,000 Å, respectively, but with the same flow rate and salt concentration. This showed that only the largest pore size, that is 2,000 Å SEC column, successfully resolved complexes in the polysome size range (*Figure 1—figure supplement 2*). To characterize the fractionation profile of the 2,000 Å SEC column and assign the resulting peaks, we injected onto the SEC column either ribosomal subunits, or polysomes, that had been previously isolated by SDG (*Figure 1A*). Using a column flow rate of 0.2 ml/min, the 40S subunit, 60S subunit, 80S monosome and a mixture of polysomes elute separately at ~50 min, 46 min, 43 min and between 33 and 41 min, respectively (*Figure 1B and C*). This showed that the first peak eluted from the SEC column contained a mixture of different n-mer polysomes, the second peak contained 80S ribosomes, the third peak contained 60S subunits, the fourth peak contained 40S subunits and the bulk of smaller protein complexes eluted in a broad peak at the end of chromatogram (*Figure 1D*). This also showed that the peak corresponding to di-somes (two ribosomes per mRNA) was also detectable (*Figure 1D*). As the 80S peak was clearly separated from the polysome peak, a P/M ratio can be calculated, similar to that achieved by SDG (*Figure 1D*). We note that the use of a 1,000 Å SEC column provided a higher resolution separation between 60S and 40S ribosomal subunits, as well as between the 40S subunit and smaller protein complexes, but resulted in the 80S peak almost overlapping the polysome peak (*Figure 1—figure supplement 2*).

To achieve a higher resolution separation, extending from larger polysomes to smaller protein complexes, in a single shot analysis, we tested the use of two SEC columns connected in series using a column flow rate of 0.2 ml/min (*Figure 1—figure supplement 3*). This showed increased resolution of di-some and tri-some peaks, along with higher, n-mer polysome complexes and improved separation of both ribosomal subunits using two 2,000 Å SEC columns in series (*Figure 1—figure supplement 3A*). A similar improved separation of polysomes was achieved by the combination of a 1,000 Å SEC column and 2,000 Å SEC column in series, while this combination also provided more effective separation in the size range from 80S to smaller protein complexes (*Figure 1—figure supplement 3B*). Fractionation of extracts using two SEC columns run in series thus provides a flexible system for efficient separation across a large range of complexes of different sizes.

Next, we optimized the column flow rate using a single 2,000 Å SEC column to minimize the total elution time required, while maintaining resolution. To do this, we chose to focus on the separation of the 80S peak from the polysome peak, given their close proximity. Using a column flow rate of 0.2 ml/min and a separation time of 60 min, the polysome peak and the 80S peak were clearly resolved (*Figure 1D* and *Figure 1—figure supplement 4*). However, at a flow rate of 1.0 ml/min, the 80S peak overlapped the polysome peak. As a compromise between analysis time and resolution, we therefore selected a flow rate of 0.8 ml/min for all subsequent experiments (*Figure 1—figure supplement 4*). Importantly, this SEC-based analysis was achieved in only 15 min, which contrasts with the hours required for SDG analysis. We also analyzed lysates individually extracted from four human cell lines (i.e. HeLa, U2OS, HCT116 p53+/+ and HCT116 p53 -/- cells) and showed that the elution profiles from the 2,000 Å SEC column were similar (*Figure 1—figure supplement 5*). We term this SEC-based uHPLC polysome fractionation method, 'Ribo Mega-SEC'.

To confirm that the first SEC peak contained polysomes, we compared the Ribo Mega-SEC chromatograms of control HeLa cell lysates with lysates that had been treated with EDTA. This treatment dissociates polysomes and 80S ribosomes into 60S and 40S subunits by chelating $Mg^{2+}$ needed for complex formation (*Klein et al., 2004*; *Nolan and Arnstein, 1969*). Using SDG analysis, we confirmed this dissociation of polysomes and 80S ribosomes following EDTA treatment. We also observed the altered (earlier) elution of each dissociated 60S and 40S subunits due to the reduced subunit density resulting from EDTA treatment (*Bengtson and Joazeiro, 2010*; *Blobel, 1971*; *Hamilton et al., 1971*; *Steitz et al., 1988*) (*Figure 2—figure supplement 1*).

We analyzed these lysates by Ribo Mega-SEC, which showed that the polysome and 80S ribosome peaks both disappeared specifically from the EDTA-treated cell lysates. There was a simultaneous appearance of two new peaks, corresponding to the 60S and 40S subunits, respectively (*Figure 2A*). These fractions elute at slightly earlier times in the presence of EDTA, given that the

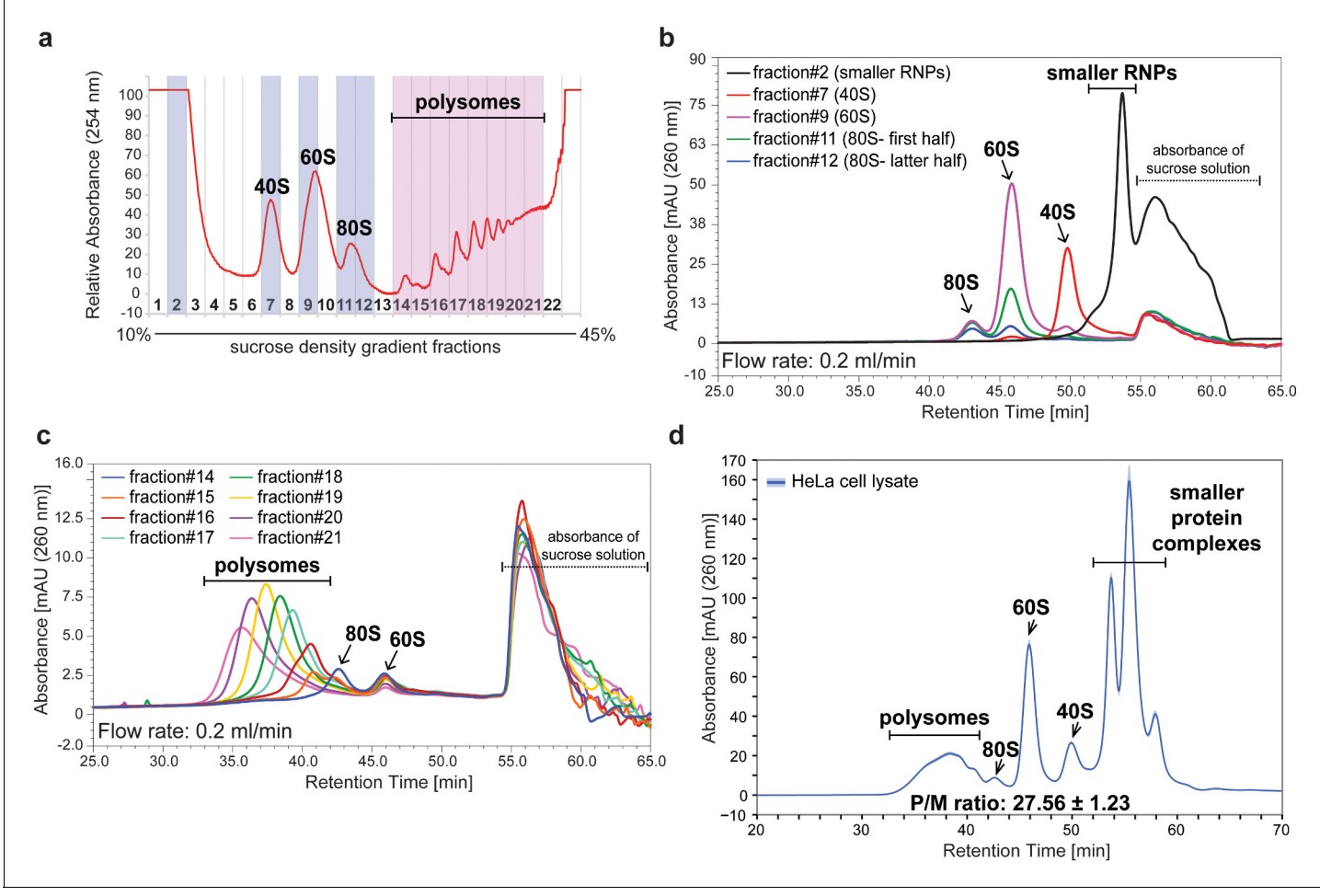

**Figure 1.** Assignment of the peaks separated by a single 2,000 Å SEC column. (a) The HeLa cell lysate containing 100 μg of RNA was separated into 22 fractions by ultracentrifugation with a 10–45% sucrose density gradient with continuous monitoring of absorbance at 254 nm. (b) Fractions highlighted in blue in (a) were analyzed by SEC with a flow rate of 0.2 ml/min using the 2,000 Å SEC column. The chromatogram of smaller RNPs, 40S, 60S, or 80S monosome is shown. (c) Fractions highlighted in pink in (a) were analyzed by SEC with a flow rate of 0.2 ml/min using the 2,000 Å SEC column. The chromatogram of each polysome fraction is shown. (d) HeLa cell lysate containing 20 μg of RNA was analyzed by SEC using a flow rate of 0.2 ml/min on the 2,000 Å SEC column. The line in the chromatogram is the mean profile and the surrounding ribbon shows the standard deviation across the three technical replicates. The retention time is indicated on the *x*-axis and the UV absorbance at 260 nm is indicated on the *y*-axis. The P/M ratio from three technical replicates was also calculated and indicated.

DOI: https://doi.org/10.7554/eLife.36530.002

The following figure supplements are available for figure 1:

**Figure supplement 1.** Comparison of the polysome profile from the lysate prepared either by Triton X-100 or CHAPS.
DOI: https://doi.org/10.7554/eLife.36530.003

**Figure supplement 2.** Comparison of the UV chromatograms of polysome profiles for three different pore size columns.
DOI: https://doi.org/10.7554/eLife.36530.004

**Figure supplement 3.** UV chromatogram of polysome profiles by sequential SEC columns.
DOI: https://doi.org/10.7554/eLife.36530.005

**Figure supplement 4.** Comparison of the separation profile by the flow rate of 0.2, 0.5, 0.8, or 1.0 ml/ min.
DOI: https://doi.org/10.7554/eLife.36530.006

**Figure supplement 5.** Comparison of Ribo Mega-SEC chromatograms for the lysates from different cell lines.
DOI: https://doi.org/10.7554/eLife.36530.007

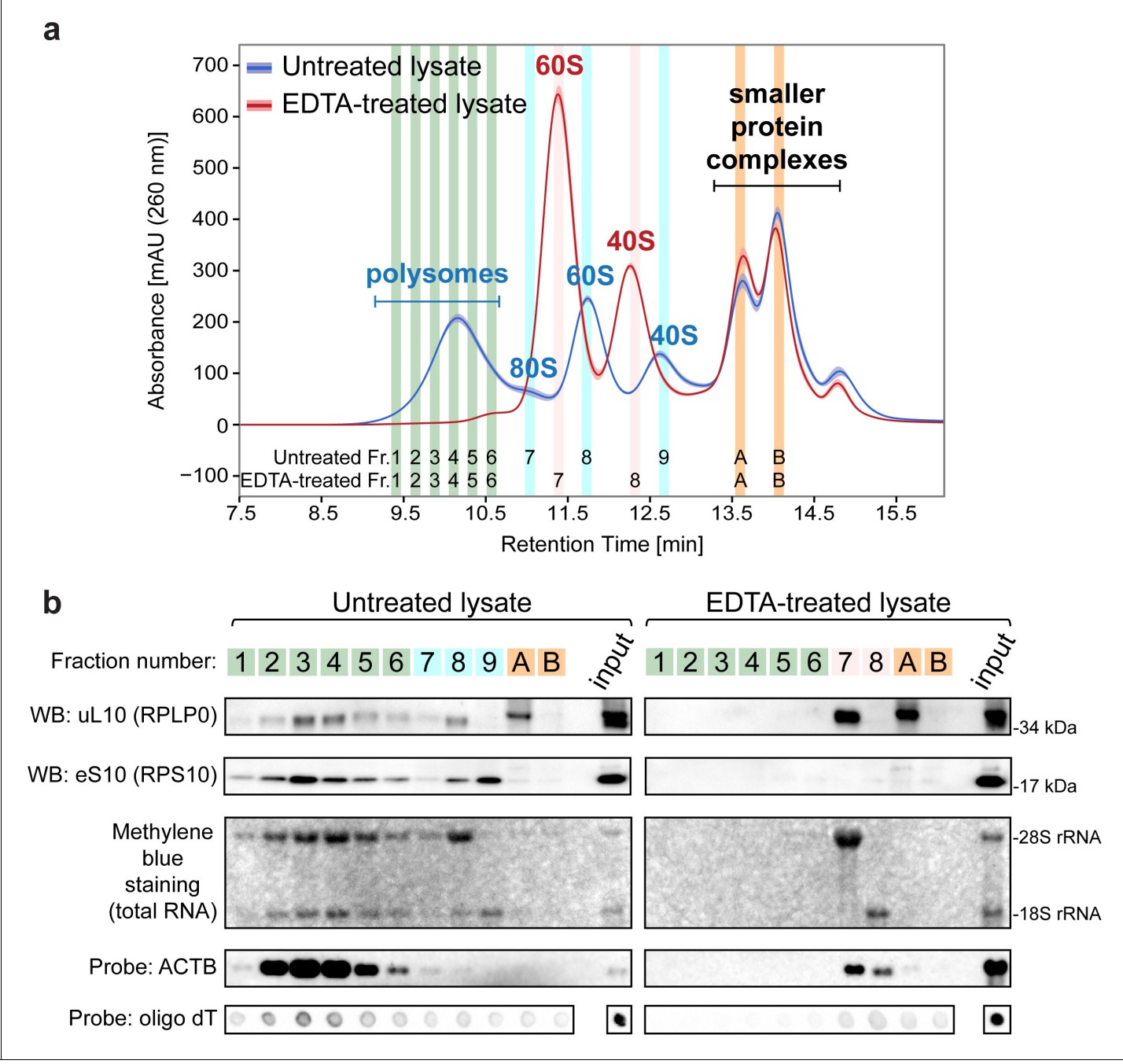

**Figure 2.** Ribo Mega-SEC chromatograms of polysomes and free ribosomal subunits. (**a**) The UV chromatogram of HeLa cell lysates either untreated or treated with 30 mM EDTA (EDTA-treated) is shown. The line is the mean profile and the surrounding ribbon shows the standard deviation across the three biological replicates. The collected fractions (Fr.) are highlighted and numbered. The retention time is indicated on the x-axis and the UV absorbance of 260 nm is indicated on the y-axis. (**b**) The fractions were analyzed either by western blotting (WB) with the indicated primary antibodies, or by northern blotting with the probes indicated at the left. Methylene blue staining visualized rRNAs. Input: 20 µg of protein and 2 µg of RNA for Methylene blue staining and detecting *ACTB* mRNA or 250 ng of RNA for detecting polyA(+) mRNA, loaded for WB and NB, respectively.

DOI: https://doi.org/10.7554/eLife.36530.008

The following figure supplements are available for figure 2:

**Figure supplement 1.** Polysome profile of untreated or EDTA-treated cell lysates by SDG analysis.
DOI: https://doi.org/10.7554/eLife.36530.009

**Figure supplement 2.** Ribo Mega-SEC chromatogram and fractions collected (*Figure 2A*).
DOI: https://doi.org/10.7554/eLife.36530.010

*Figure 2 continued on next page*

*Figure 2 continued*

**Figure supplement 3.** Western blotting with anti-eS10 antibody across the fractions separated from EDTA-treated cell lysates.

DOI: https://doi.org/10.7554/eLife.36530.011

density of each ribosomal subunit was decreased, and their effective size increased, following EDTA treatment, as noted above (*Gesteland, 1966*).

We next collected 48 fractions, ranging from polysomes to smaller protein complexes (*Figure 2— figure supplement 2*) and characterized the SEC peaks further, using both western and northern blotting to compare the compositions of specific fractions (*Figure 2A* and *Figure 2—figure supplement 2*). The polysome fractions (fractions 1–6) in the untreated control lysate contained RPs, that is, uL10 (RPLP0) and eS10 (RPS10), and 28S and 18S rRNAs. The polysome peak was also highly enriched in PolyA(+) mRNA. For example, beta actin mRNA (*ACTB*) was enriched specifically in the polysome peak (*Figure 2B*). Moreover, the 60S and 40S subunit peak fractions were also enriched in the corresponding 60S and 40S-specific RPs and rRNAs. As expected, these RP, rRNA and mRNA signals were significantly reduced in the peak polysome fractions from EDTA-treated cell lysates (*Figure 2B*). Following EDTA treatment, uL10 accumulated in two fractions; that is, a dissociated 60S subunit fraction (fraction 7) and the fraction not co-eluted with 28S rRNA (fraction A). eS10 was neither detected in the dissociated 40S subunit in the fractions analysed by Ribo Mega-SEC, nor in the dissociated 40S fractions analysed by SDG, but was detected instead in the fractions containing smaller protein complexes (*Figure 2B* and *Figure 2—figure supplements 1* and *3*). This suggests that eS10 is one of the RPs released from ribosomal subunits after EDTA treatment, as reported previously for the 5S RNP (*Blobel, 1971*; *Steitz et al., 1988*).

We next examined the stability and activity of polysomes obtained by Ribo Mega-SEC, during sequential separations and high salt treatment. First, we tested polysome stability over time by taking three fractions in the polysome peak (highlighted fractions 1–3 in *Figure 3A*) and analyzing each fraction individually, both by a second round of Ribo Mega-SEC and by SDG analysis (*Figure 3B*). Earlier SEC fractions migrated in the heavier polysome region (>5 ribosomes per polysome) by SDG, and later SEC fractions migrated in the lighter polysome region (~3 ribosomes per polysome) by SDG. In addition, no dissociated ribosomal subunits from the Ribo Mega-SEC purified polysomes were detected by either the SEC, or SDG approaches. This demonstrates that the polysomes separated by Ribo Mega-SEC are stable and remain intact (*Figure 3B*).

We also analyzed two sets of polysome fractions isolated by Ribo Mega-SEC using electron microscopy (EM); that is (a) material from earlier eluting polysome fractions (EM: a in *Figure 3A*) and (b) material from an intermediate eluting polysome fraction (EM: b in *Figure 3A*). The complexes from each fraction were applied directly onto an EM grid, without dialysis, and viewed in a JEOL 2010F with FEG (Field Emission Gun) electron microscope in TEM mode at a nominal magnification of 40,000X. This EM analysis showed that earlier eluting SEC fractions contain larger polysomes, with elongated and heterogeneous shapes (*Figure 3C*). These may correspond to 'HMW and/or [HMW]$^n$ ($n$ = 2) polysomes (i.e. assemblies of polysomes)' (*Viero et al., 2015*). In contrast, polysomes composed of four ribosomes accounted for a major portion of the intermediate eluting SEC fraction and their shapes were more round and homogeneous (*Figure 3B and C*). These may correspond to 'rounded' polysomes (*Viero et al., 2015*). In addition, some larger polysomes were also found in this fraction, which is consistent with the data obtained by SDG analyses above (*Figure 3B and C*).

In a second round of stability tests, we examined SDG and SEC profiles of HeLa cell extracts prepared and separated under either normal salt concentrations, or high-salt concentrations. Using SDG analysis, we detected a decrease in intensity of the 80S peak and accumulation of free 60S and 40S subunits. This is likely due to vacant 80S couples dissociating into free subunits in the high-salt buffer (*Figure 3—figure supplement 1A*) (*Martin and Hartwell, 1970*; *van den Elzen et al., 2014*). We also detected a reduction of both the amount and resolution of polysome peaks in the SDG separations due to the increase of salt concentration (*Figure 3—figure supplement 1A*) (*Strnadova et al., 2015*). We then analyzed these same lysates by SEC, which also showed a similar decrease of polysome peak abundance (*Figure 3—figure supplement 1B*). Based on our observations that, under high-salt concentrations, (i) each peak analyzed by SDG elutes earlier and (ii) each peak analyzed by SEC elutes later, we infer that high salt reduces the size of ribosomal subunits,

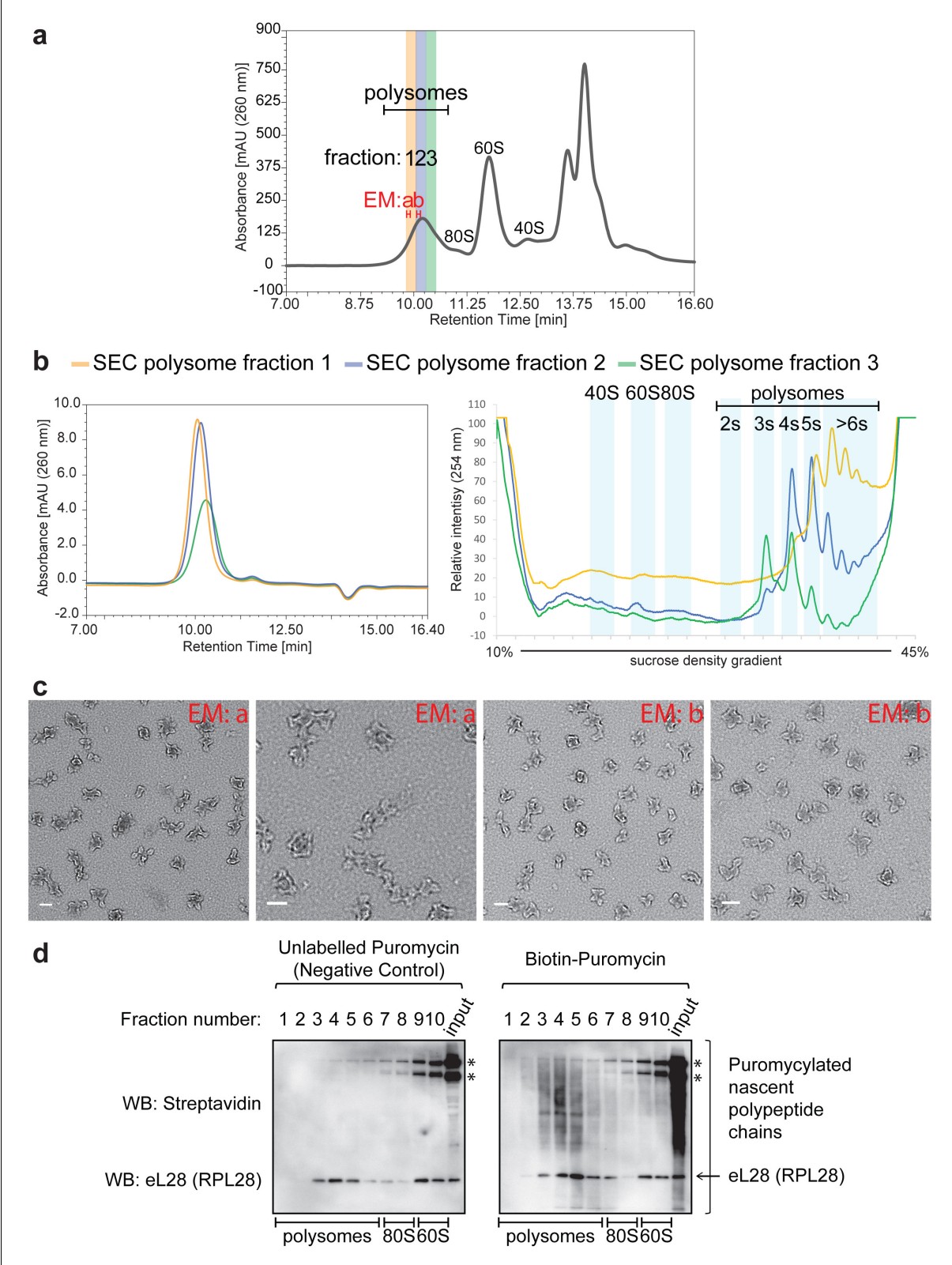

**Figure 3.** Stability of polysomes isolated by Ribo Mega-SEC. (**a**) Fractions analyzed by the subsequent SDG analysis are numbered and highlighted in colors (yellow, blue, and green) in the Ribo Mega-SEC profile of HeLa cell lysate. Fractions analyzed by the subsequent EM analysis are termed as 'EM a' and 'EM b'. The retention time is indicated on the *x*-axis and the UV absorbance of 260 nm is indicated on the *y*-axis. (**b**) Each fraction in (**a**) was analyzed by ultracentrifugation with a 10% to 45% sucrose density gradient. The regions where ribosomal subunits (40S, 60S and 80S) and the number

*Figure 3 continued on next page*

*Figure 3 continued*

of ribosomes (2 s, 3 s, 4 s, 5 s and >6 s) in polysomes sediment are labeled. (**c**) Fractions labeled as 'EM a' and 'EM b' in (**a**) were analyzed by EM analysis. Scale bar showing 100 nm is indicated at the bottom left in each image. (**d**) Each fraction in *Figure 3—figure supplement 2* was incubated with either unlabelled puromycin (Negative control), or biotin-labelled puromycin. The biotin-puromycylated nascent polypeptide chains and eL28 (RPL28) were detected by WB with streptavidin and anti-RPL28 antibody. *endogenous biotinylated proteins. Input: 20 µg of protein in lysates incubated with either unlabeled puromycin, or biotin-labeled puromycin was loaded.

DOI: https://doi.org/10.7554/eLife.36530.012

The following figure supplements are available for figure 3:

**Figure supplement 1.** Comparison of polysome profiles of lysates extracted either by normal salt or by high-salt containing buffer.

DOI: https://doi.org/10.7554/eLife.36530.013

**Figure supplement 2.** Ribo Mega-SEC profile for *in vitro* puromycylation (*Figure 3D*).

DOI: https://doi.org/10.7554/eLife.36530.014

probably because of a loss of ribosomal proteins and/or proteins associated with ribosomal subunits.

Next, we examined whether the ribosomes in polysome fractions separated by Ribo Mega-SEC retained peptidyl transferase activity, which would provide a clear indicator of stability and retention of function after fractionation. To do this, we employed *in vitro* puromycin labeling (*Figure 3D* and *Figure 3—figure supplement 2*) (*Aviner et al., 2013*). As was true for all experiments in this study, we used lysates from cells treated with cycloheximide for this analysis. This was possible because short-term treatment of cells with cycloheximide has no significant effect on nascent polypeptide chain puromycylation (*David et al., 2012*). We detected nascent polypeptide chains linked with bio-tin-labeled puromycin specifically in the polysome fractions (*Figure 3D*). A streptavidin-HRP signal was not observed in the 60S subunit fractions, or when extracts were treated with unlabeled-puromycin (negative control) (*Figure 3D*). These data show that, using Ribo Mega-SEC, both intact and translation-active polysomes can be resolved from cell extracts efficiently (~11 min after injection).

An important distinction between density-gradient-based fractionation and uHPLC-based separation is the inherent improvement in reproducibility through the use of automated injection and fraction-collection systems. Many fields, including biochemistry and pharmacology, rely on the reproducible retention times and quantitation provided by automated uHPLC systems. We have evaluated reproducibility here for Ribo Mega-SEC through the analysis of three biological replicates of either untreated, or EDTA-treated, cell lysates. Statistical comparison of these chromatograms showed very high Pearson correlation coefficients of ~0.99 across the biological replicates (*Figure 4A* and *Figure 4—figure supplement 1*). Polysome profiles generated by SDG analysis from three biological replicates of untreated cell lysates also showed high Pearson correlation coefficients, but consistently lower than those from Ribo Mega-SEC (*Figure 4B*). Moreover, we found an ~5 to 10 s difference (equivalent to 80 µl to 160 µl difference) between the SDG replicates in the polysome region, possibly due to the variability in density of the sucrose gradients in each tube (*Figure 4C*). These data show that the Ribo Mega-SEC approach is highly reproducible and compares favourably in this regard with polysome isolation using SDG.

We next took advantage of the optimized Ribo Mega-SEC method to examine whether we detect differences in the relative levels of polysomes and ribosomal subunits when extracts are analyzed from HeLa cells responding to growth under conditions of amino acid starvation. This showed dramatically decreased polysome levels in the lysates from amino acid starved cells, with a corresponding distinct increase in the peaks for 80S ribosome, and 60S and 40S subunits, consistent with previous reports (*Caldarola et al., 2004*) (*Figure 5A*). Quantitation of the peak areas and the P/M ratio confirmed the expected global reduction of translation in response to amino acid starvation (*Figure 5B*).

Translation of 5' TOP mRNAs has been reported to be repressed by amino acid starvation, due to the release of this class of mRNAs from polysomes (*Damgaard and Lykke-Andersen, 2011*). Our data here using Ribo Mega-SEC support these observations on the loss of 5' TOP mRNAs from polysomes after starvation. Thus, we see that the *uL23 (RPL23a)* mRNA, a 5' TOP mRNA, was specifically decreased in the polysome fractions of extracts isolated from amino acid starved cells, while the *ACTB* mRNA mostly remained associated with polysomes, although we detected a small portion of *ACTB* mRNA also redistributed into the ribosomal subunit fractions (*Figure 5C*). These data indicate

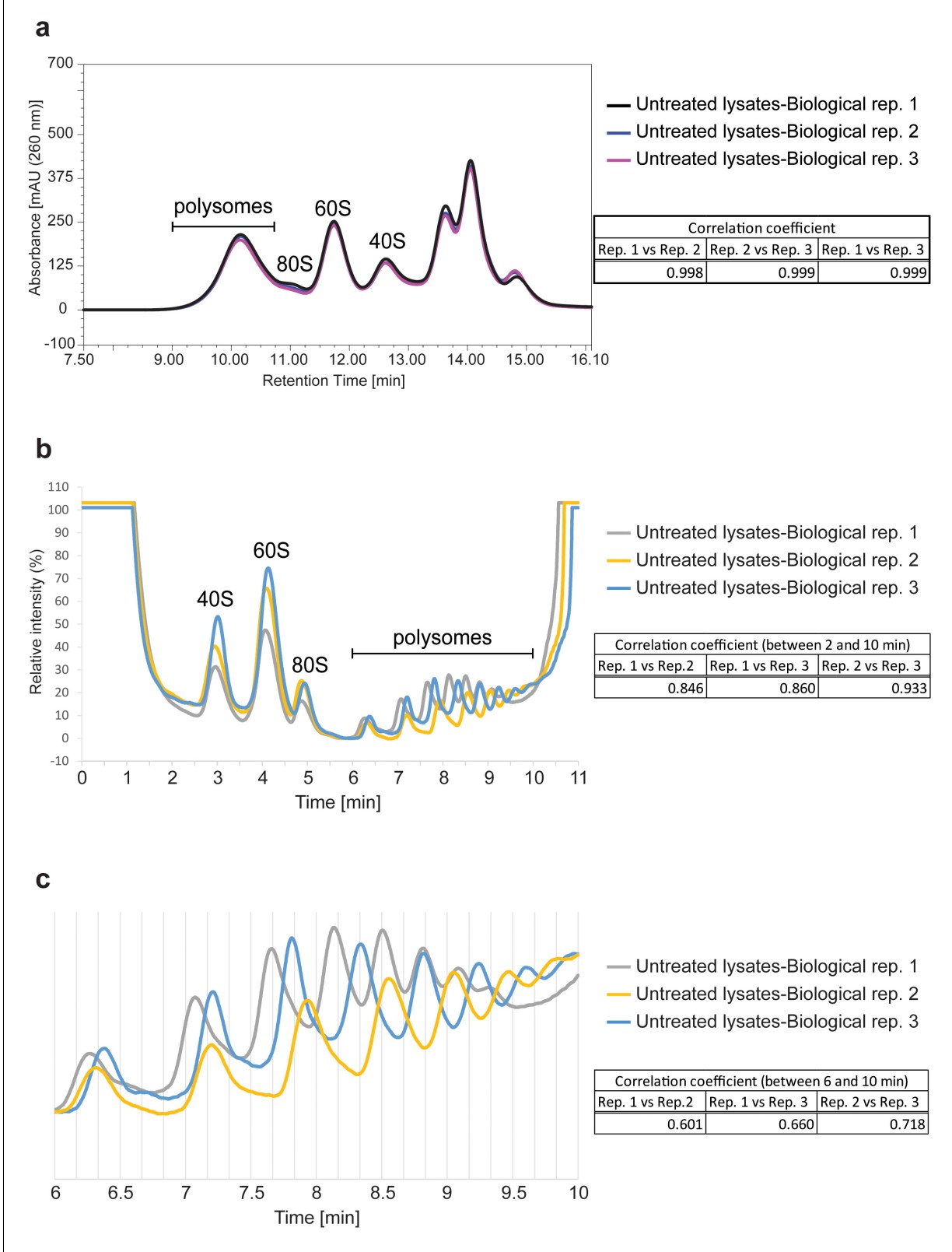

**Figure 4.** Reproducibility of Ribo Mega-SEC and SDG analysis. (**a**) The UV chromatograms of Ribo Mega-SEC from the three biological replicates of untreated cell lysates were showed. The retention time is indicated on the *x*-axis and the UV absorbance of 260 nm is indicated on the *y*-axis. The Pearson correlation coefficient calculated for the three biological replicates between 8.0 min and 16.1 min (replicate 1 versus replicate 2, replicate 1 versus replicate 3, and replicate 2 versus replicate 3) was also indicated at the right. (**b**) The polysome profiles of SDG analysis from the three biological

*Figure 4 continued on next page*

Figure 4 continued

replicates of untreated cell lysates are shown. The time is indicated on the *x*-axis and the UV absorbance of 254 nm is indicated on the *y*-axis. The Pearson correlation coefficient calculated for the three biological replicates between 2.0 min and 10.0 min (replicate 1 versus replicate 2, replicate 1 versus replicate 3, and replicate 2 versus replicate 3) is also indicated at the right. (c) The polysome region from 6.0 min to 10.0 min in (b) was expanded and the Pearson correlation coefficient was calculated between these time points.

DOI: https://doi.org/10.7554/eLife.36530.015

The following figure supplement is available for figure 4:

**Figure supplement 1.** The three biological replicates showing elution profiles of EDTA-treated cell lysates by Ribo Mega-SEC.

DOI: https://doi.org/10.7554/eLife.36530.016

that the Ribo Mega-SEC approach is well suited for use in studies to characterize changes in translational activity in cells, either under different growth conditions, or in response to drugs, physical, or genetic perturbations.

The previous examples involved analysis of polysomes in extracts isolated from human cell lines. We also evaluated how effectively Ribo Mega-SEC can be used to analyse polysomes in extracts prepared from mammalian tissue samples. In addition, we wished to test whether this approach can be used to perform MS-based proteomic analyses on the isolated polysome fractions. To do this, we applied Ribo Mega-SEC to analyze the proteins associated with polysomes and ribosomal subunits in mouse liver tissue. Extracts were prepared from mouse liver and fractionated by SEC, which showed a similar pattern of polysome separation as seen with extracts prepared from HeLa cells, also with very high Pearson correlation coefficients (~0.99) across biological replicates (*Figure 6—figure supplement 1*). This showed that translation complexes in liver tissue are mostly associated with polysomes, rather than 80S monosomes.

Next, we optimized the SEC running buffer to facilitate downstream proteomic analysis of the fractionated polysomes. For this, we found it was important to omit heparin from the SEC running buffer, as it interferes with MS analysis and is difficult to remove after SEC fractionation using conventional proteomics sample preparation techniques. We confirmed that the separation profiles were similar using SEC running buffer either with, or without, heparin, and also confirmed that polysomes and ribosomal subunits remained intact under both conditions (*Figure 6—figure supplement 2*). We then collected 24 SEC fractions, spanning the separation range from polysomes to smaller complexes (*Figure 6A and B*). Each fraction was digested with trypsin and the resulting peptides cleaned to remove the buffer components, prior to LC-MS/MS analysis on a QExactive plus mass spectrometer (*Figure 6A*) (*Larance et al., 2016*; *Larance et al., 2015*; *Ly et al., 2015*; *Ly et al., 2017*).

The data from two biological replicates yielded >58,800 unique peptides, which were mapped to 5,158 protein groups, each identified with two or more peptides per protein. We used iBAQ intensity for label-free quantitation to estimate protein abundance and compare protein abundance across the fractions (*Figure 6C and E* and *Figure 7*) (*Larance et al., 2016*).

We further analyzed the data to identify both polysome-associated and ribosomal subunit-associated proteins, using hierarchical clustering across the 24 fractions. To determine the number of protein clusters required, we tested the generation of 1–2,000 separate clusters from our dataset and calculated the mean Pearson correlation within each cluster (*Figure 6—figure supplement 3A and B*) (*Kirkwood et al., 2013*). We picked 400 protein clusters as an optimum, because this showed a high mean correlation coefficient (~0.95) for each cluster, while minimizing the subdivision of known protein complexes between different clusters (*Figure 6—figure supplement 3A*, *Supplementary file 1*). The large and small ribosomal subunit proteins predominantly clustered separately, with cluster 197 containing 39 proteins from the 60S subunit (*Figure 6C and D*). Cluster 454 contained 30 proteins, mostly from the 40S subunit, including the known accessory proteins Gnb2l1 (*Link et al., 1999*) (*Figure 6E and F*). Moreover, two proteins (Slc2a2 and Myl12a) were observed in the 40S cluster as candidate 40S-associated factors (*Figure 6E and F*).

Using mouse liver tissue, we detected in total 78 mouse RPs across the fractions, of which 33 were small ribosomal subunit proteins and detected in the polysomes/80S and 40S subunit fractions. The other 45 were large ribosomal subunit proteins and detected in polysomes/80S and 60S subunit fractions (*Figure 7*) (*Nakao et al., 2004*). We also detected multiple translation initiation factors (eIFs), which were mainly in 40S fractions, forming the 48S initiation complex. We also found that

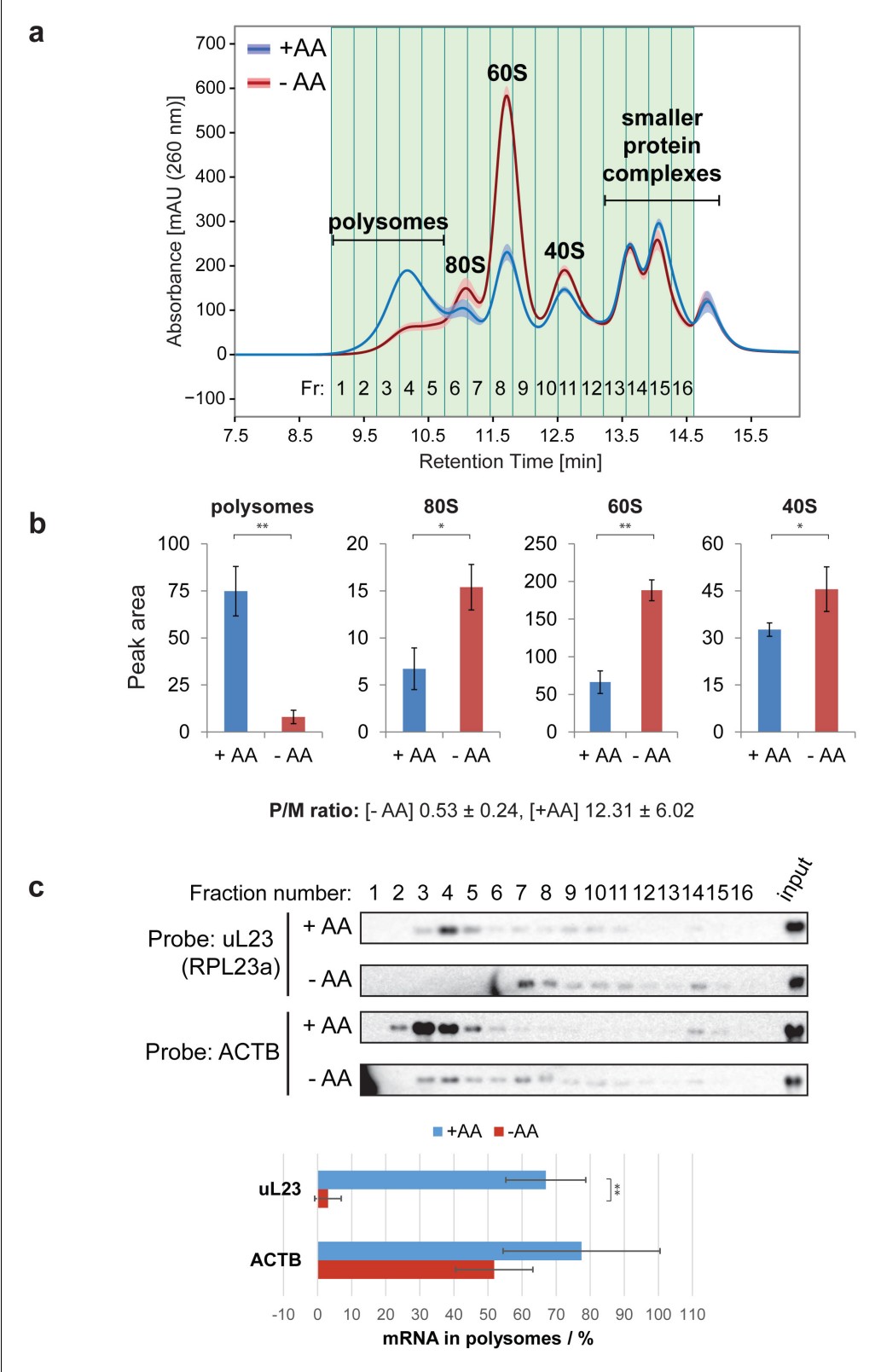

**Figure 5.** Ribo Mega-SEC profile with or without amino acid starvation. (**a**) Sixteen fractions separated from the lysates of HeLa cells grown in EBSS, either with (+AA), or without (-AA), exogenous amino acids for 2 hr, were collected. The line is the mean profile and the surrounding ribbon shows the standard deviation across the three biological replicates. The collected fractions (Fr.) are highlighted and numbered. The retention time is indicated on the *x*-axis and the UV absorbance of 260 nm is indicated on the *y*-axis. (**b**) The area of each peak of either polysomes, or free ribosomal subunits,
*Figure 5 continued on next page*

*Figure 5 continued*

was quantified and graphed. The values are mean (±standard deviation) of three biological replicates. *p<0.05, **p<0.01. The P/M ratio from three biological replicates was also calculated and indicated. (c) The fractions were analyzed by northern blotting with the probes indicated at the left. Input: 2 μg of RNA from the lysate was loaded. Quantifications of the fraction (in percent of the total amount) of uL23 and ACTB mRNAs found in the polysome fractions were performed and the graphs are shown at the bottom. The values are mean (±standard deviation) of three biological replicates. **p<0.01.

DOI: https://doi.org/10.7554/eLife.36530.017

Poly(A) binding proteins 1 and 4 (PABPC1, PABPC4) were enriched in the polysome fractions, with a small portion in the fractions containing smaller protein complexes (*Schäffler et al., 2010*). Proteins from the Exon Junction Complex (EJC - EIF4A3, RBM8A, RNPS1, MAGOHB, CASC3) were detected specifically in the polysome fractions and ribosomal subunit fractions, consistent with a previous report showing that EJC proteins eluted in the void volume of a gel filtration column (>2 MDa) (*Singh et al., 2012*). We also detected UPF1, a protein involved in nonsense-mediated decay of mRNA, mainly in polysome fractions, as well as in 40S fractions (*Min et al., 2013*). In addition, we detected translation elongation factors (eEFs) and translation termination factors (eRFs), which were present in polysomes and 80S fractions, although the majority of these proteins were in the SEC fractions containing smaller complexes (*Figure 7*), consistent with previous studies (*Eyler et al., 2013*; *Sivan et al., 2011*; *Sivan et al., 2007*).

In summary, we conclude that Ribo Mega-SEC provides an efficient and flexible approach for isolating polysomes and ribosomal subunits that is compatible with downstream proteomic analysis on the isolated material. The method can be applied to study extracts from both cell lines and tissue samples and thereby provides a convenient new technology for characterizing polysome-associated proteins in cells under different growth and response conditions.

To maximize community access to the entire dataset described in this study, the proteomics data, including raw MS files and MaxQuant-generated output, are freely available via the ProteomeXchange PRIDE repository with the data set identifier PXD008913.

## Discussion

The Ribo Mega-SEC method described here provides a powerful new approach for enhancing the analysis of gene expression at the level of active translation complexes. We have shown that Ribo Mega-SEC provides a rapid and reproducible method for the isolation of polysomes and ribosomal subunits using uHPLC and is therefore an accessible alternative to the use of conventional SDG methods. In particular, the Ribo Mega-SEC approach is well suited to broadening access for researchers to analyze polysomes more widely in future studies on regulatory responses and disease mechanisms in mammalian cells and tissues.

Polysome analysis is currently performed almost exclusively using SDGs. This is a well-established and effective method but also is widely recognized to have some limitations. For example, it is comparatively slow to carry out, involving a long ultracentrifugation step, and requires specialist equipment and expertise to use reliably. In addition, the multiple handling steps involved in setting up SDG analyses, together with the subsequent fraction collection and analysis steps, all introduce potential sources of experimental and technical variation that overall can reduce detailed reproducibility of gradient profiles between SDG experiments, as previously recognised (*Ingolia et al., 2009*). Another practical issue arises from the sensitivity of the detectors commonly used for analyzing SDG fractions and the requirement for establishing a consistent signal to noise baseline. This can potentially complicate the accurate comparison of SDG polysome profiles between experiments, without additional normalization steps, particularly when analyzing samples with low RNA amounts (*Figure 2—figure supplement 1*). Moreover, due to the high viscosity of the sucrose solution used for SDG, either further purification and/or dialysis is needed if the separated polysomes and ribosome subunits are to be used for subsequent experiments, such as Cryo-EM analysis (*Grassucci et al., 2007*). In combination, all these issues outlined above have limited the wider application of polysome analyses for many types of gene expression studies addressing regulatory responses and disease mechanisms in human cells and model organisms.

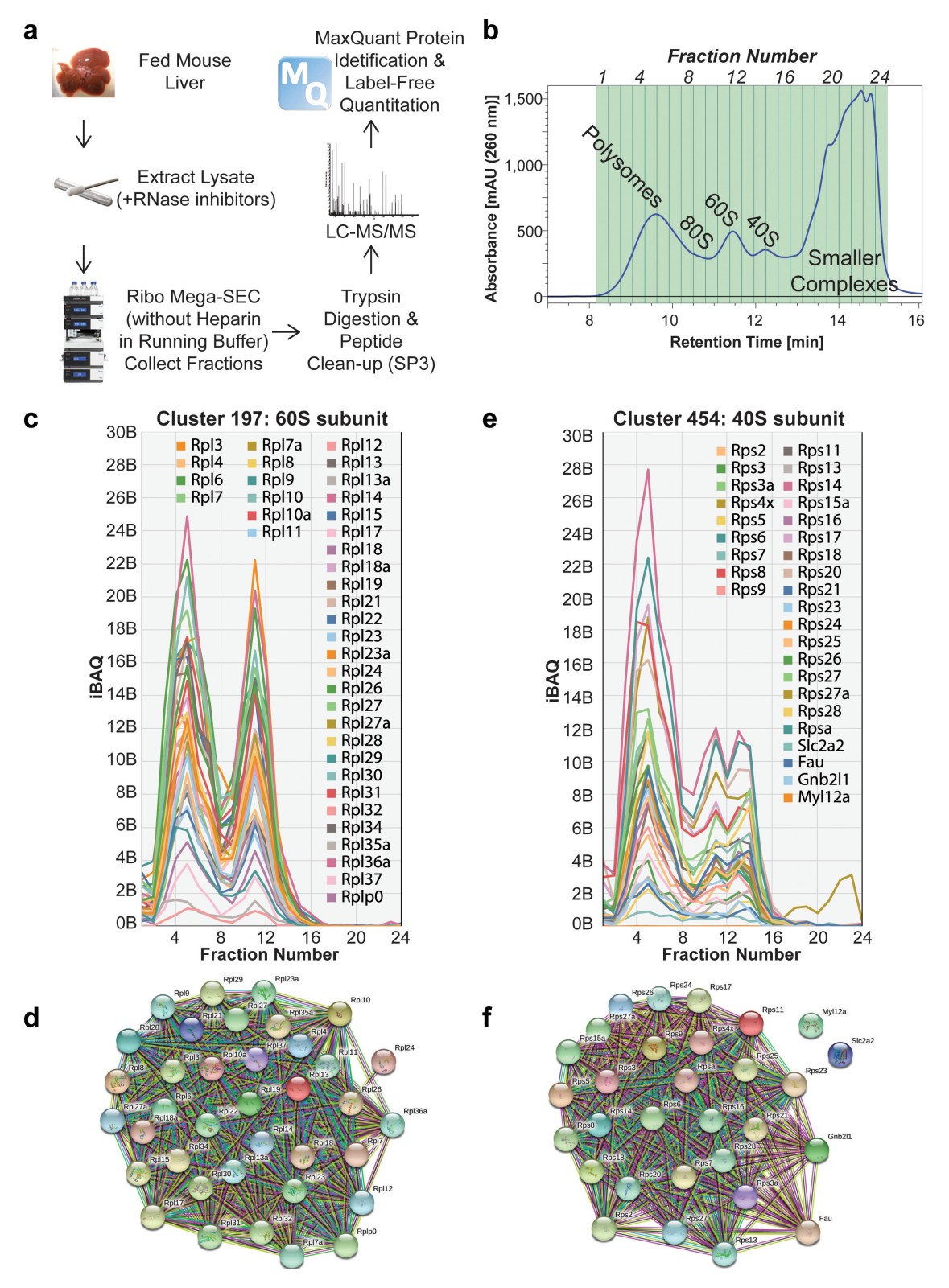

**Figure 6.** Proteomic analysis of Ribo Mega-SEC fractions from mouse liver tissue. (**a**) Workflow for Ribo Mega-SEC analysis of mouse liver tissue and LC-MS/MS methodology. (**b**) Ribo Mega-SEC profile of mouse liver tissue. The collected fractions are highlighted and numbered at the top of the chromatogram. The retention time is indicated on the *x*-axis and the UV absorbance of 260 nm is indicated on the *y*-axis. (**c**) Ribo Mega-SEC elution profiles for all proteins identified in example cluster 197 were displayed as line graphs with the mean iBAQ intensity (*y*-axis) plotted against each elution

*Figure 6 continued on next page*

*Figure 6 continued*

fraction (*x*-axis). (**d**) STRING interaction network for proteins identified in example cluster 197. (**e**) Ribo Mega-SEC elution profiles for all proteins identified in example cluster 454 were displayed as line graphs with the mean iBAQ intensity (*y*-axis) plotted against each elution fraction (*x*-axis). (**f**) STRING interaction network for proteins identified in example cluster 454.

DOI: https://doi.org/10.7554/eLife.36530.018

The following figure supplements are available for figure 6:

**Figure supplement 1.** The three biological replicates of the elution profile of mouse liver extract analysed by Ribo Mega-SEC.

DOI: https://doi.org/10.7554/eLife.36530.019

**Figure supplement 2.** Comparison of Ribo Mega-SEC profiles using a mobile phase with or without heparin.

DOI: https://doi.org/10.7554/eLife.36530.020

**Figure supplement 3.** Hierarchical clustering of protein elution profiles in the mouse liver tissue.

DOI: https://doi.org/10.7554/eLife.36530.021

The Ribo Mega-SEC approach we introduce here overcomes a number of the limitations associated with polysome analysis by SDG for studies on mammalian cells and tissues. For example, Ribo Mega-SEC involves a single, rapid uHPLC step; the process from injecting lysate to collecting fractions can be completed within 15 min, while keeping the sample at 4–5°C through all steps. This has obvious benefits for biochemical analyses. Moreover, the use of uHPLC provides exceptional reproducibility of retention time between experiments, due to the combination of automated, highly accurate sample injection, an automatic UV signal zeroing at the start of each sample run and high quality, robust components (pumps/valves) and columns. We also demonstrate that Ribo Mega-SEC can be used in conjunction with physiological buffers that are compatible with most subsequent analytical techniques, including MS-based proteomics, EM analysis and biochemical assays, for example translation activity assays (*Figures 3*, *6* and *7*). Ribo Mega-SEC profiles can thus reflect the physiological status of polysomes and ribosomes *in vivo*, and the data they provide indicate that mRNA translation occurs predominantly in polysome complexes (as compared to 80S 'monosome' complexes) (*Aspden et al., 2014*; *Noll, 2008*; *Warner and Knopf, 2002*) in mouse liver and in human cancer cell lines.

Using a single uHPLC instrument, as many as 48 different samples can be automatically separated and fractionated by Ribo Mega-SEC within 16 hr. Ribo Mega-SEC is therefore suited for high-throughput experiments and for combining with techniques such as RNA-Seq analysis and proteomic analysis (*Figures 6* and *7*). We have also shown that the separate fractions in a single polysome peak have different sized n-mer polysomes (*Figures 1* and *3*), and these fractions as well as the ribosomal subunit fractions can also be re-analyzed and further separated by a second round of SEC. We note that, thanks to the rapid separation provided by uHPLC, even performing two sequential rounds of Ribo Mega-SEC analysis to achieve higher resolution separation of both different sizes of polysomes and ribosomal subunits can be achieved comfortably within 30–45 min total analysis time. This is faster than performing a single round of SDG fractionation, as well as delivering the polysomes and ribosomal subunits already in a physiological buffer.

The specific setup and conditions used for Ribo Mega-SEC fractionation can be customized in different experiments to optimize the analysis depending on whether speed, or resolution is most critical. For example, by employing a lower flow rate (e.g. 0.2 ml/min), higher resolution separation of large complexes can be achieved (*Figure 1—figure supplement 4*). Alternatively, employing two SEC columns connected in series, with a flow rate of 0.2 ml/min, provides a higher resolution separation of complexes from larger polysomes to smaller protein complexes in a single shot SEC analysis (*Figure 1—figure supplement 3*). In this case, the trade-off is that a longer separation time is required. For many experiments, therefore, using a single SEC column with a faster 0.8 ml/min flow rate, is optimal. The pore size of the SEC column can also be varied to help customize separation for specific applications. For example, we show that using a 1,000 Å SEC column with a flow rate of 0.2 ml/min improves the separation of 60S and 40S ribosomal subunits as well as 40S subunits from smaller protein complexes (*Figure 1—figure supplement 2*). This can be helpful for studying translation initiation steps. There is a clear scope for enhancing further the specific application of Mega-SEC fractionation methods based upon varying these parameters such as pore size, buffer, number and type of columns used in series and flow rates etc.

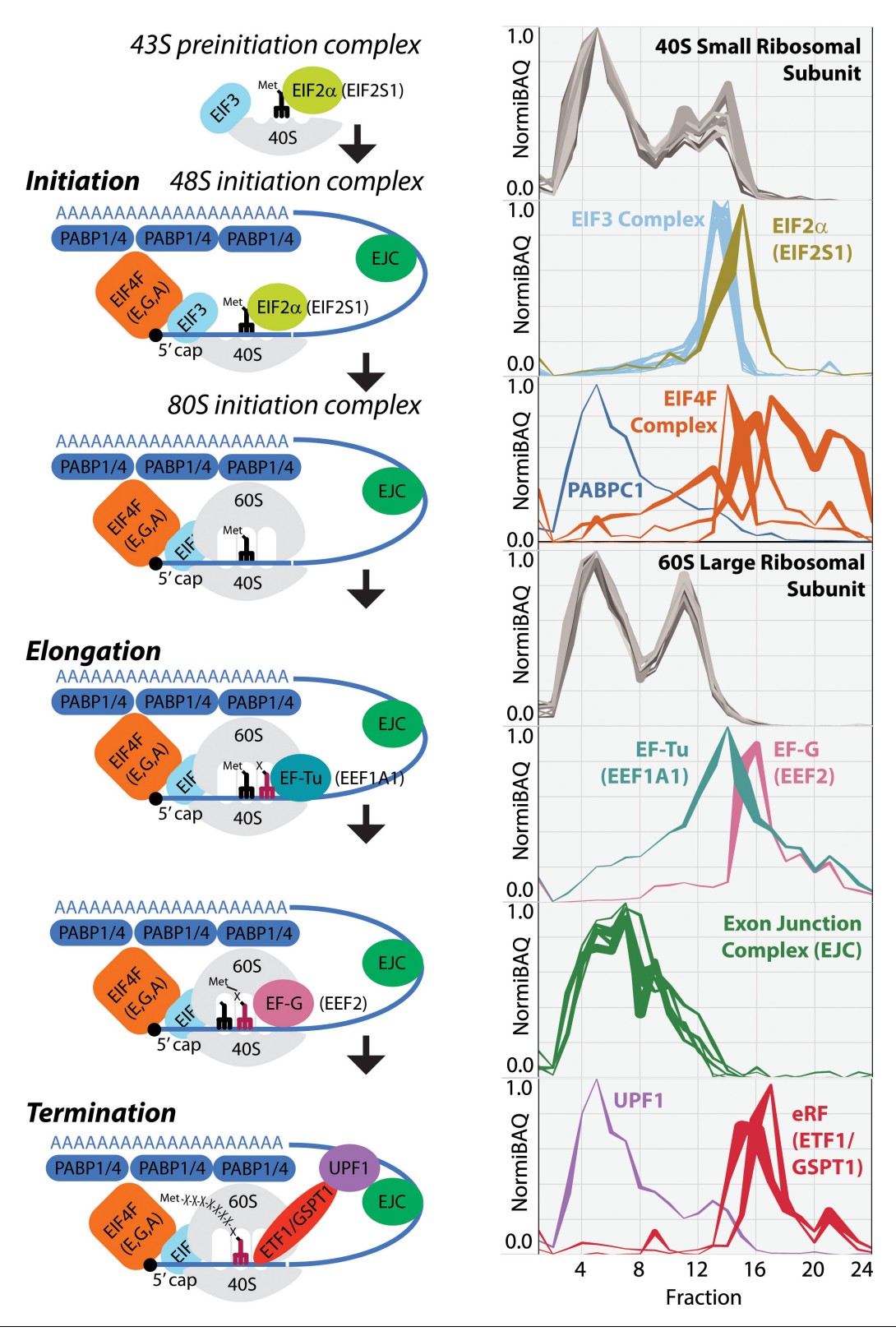

**Figure 7.** Schematic image of whole translation process highlighted with the identified proteins in mouse liver tissue. Ribo Mega-SEC elution profiles for 40S small ribosomal subunit, EIF2alpha, EIF3 complex, EIF4F complex, PABPC1, 60S large ribosomal subunit, EF-Tu, EF-G, exon junction complex (EJC), UPF1, and eRF are displayed as line graphs with the mean normalized iBAQ intensity ± the standard deviation (y-axis) plotted against each elution fraction (x-axis).

*Figure 7 continued on next page*

*Figure 7 continued*

DOI: https://doi.org/10.7554/eLife.36530.022

Polysome profiling is now a popular technique. It is used, for example, to assess changes in translation efficiency by examining a pool of efficiently translated mRNAs associated with more than three ribosomes, referred to as a 'heavy polysome', typically involving isolation of the heavy polysome by SDG (*King and Gerber, 2016*; *Piccirillo et al., 2014*). Ribosome profiling is another popular technique used to assess the changes in translation efficiency by identifying and quantifying the ribosome-protected fragments by RNAseq (*Ingolia et al., 2012*; *Ingolia et al., 2009*). However, polysome profiling offers technical advantages, because ribosome profiling can be biased toward the identification of abundant mRNAs that show large changes (*Masvidal et al., 2017*). A recent report has described an efficient method to enrich heavy polysomes in ~1 ml, using a non-linear sucrose gradient, which helps to overcome one of the issues when larger volumes are collected across several polysome fractions after a typical SDG analysis. Nonetheless, this method still involves the preparation of sucrose gradients and a long ultracentrifugation step (*Liang et al., 2018*).

The Ribo Mega-SEC approach shows that the heavy polysomes are enriched in the first half of the polysome peak, eluted between 9 and 10.25 min after injecting the lysates (*Figure 3A–C*), which is equivalent to ~1 ml in volume. However, it is thought that the bulk of 80S monosomes are involved in translation elongation. Therefore, pre-selection of heavy polysomes may bias the interpretation of an *in vivo* translation activity (*Heyer and Moore, 2016*). A more comprehensive picture of the translational flux *in vivo* may be provided therefore by isolating and quantifying mRNAs from fractions containing both polysomes and the 80S monosome, which can be collected within ~11.25 min of starting analysis using the Ribo Mega-SEC method (*Figure 2A*). We therefore suggest that Ribo Mega-SEC offers significant technical and practical advantages for future studies on post-transcriptional gene expression in mammalian cells and tissues involving the isolation of active translation complexes.

While we have demonstrated clear advantages provided by Ribo Mega-SEC, no single method is best suited to every situation and we acknowledge two potential features of the Ribo Mega-SEC method that may be relevant to whether it is used for specific applications. The first is that polysomes are predominantly eluted by a single column SEC in a single major peak, albeit distributed across multiple fractions. We show here that this resolution of polysomes can be improved by running two SEC columns in series in a single shot experiment. Nonetheless, if the separation of different higher order n-mer polysome species is critical for a specific experiment, it may still be preferable to perform SDG analysis, either instead of, or in addition to, Ribo Mega-SEC. However, while analysis of mRNAs with varying numbers of ribosomes is reported for 'TrIP-Seq' or 'Poly-Ribo-Seq' using SDG (*Aspden et al., 2014*; *Floor and Doudna, 2016*), our analyses indicate that the isolation of different polysome size classes by SDG might underestimate the true level of cross-contamination between fractions because of the increased baseline in the polysome peak region, which indicates that polysome 'peaks' are not homogeneous. Inevitably, in addition to the major species in the peak, they also contain mixtures of polysomes of different sizes. A second issue we have identified is the assessment of vacant 80S couples by high-salt treatment (*Figure 3—figure supplement 1*). This arises because SEC separation is sensitive to changes in mobile phase components, such as salts. However, we note this use of high-salt extracts represents a special case that is not involved in most routine studies of polysomes in cells and tissues. Instead, fractionation using high salt may be most relevant to specific biochemical studies, for example on the detailed mechanism of ribosome assembly. Therefore, the choice of whether best to use either conventional SDG methods, or the new Ribo Mega-SEC method, can be made by researchers in light of the required experimental design and mindful of the specific pros and cons of each approach. In many cases, it may be sensible to analyze extracts using both methods, as illustrated here.

In summary, we have demonstrated that the Ribo Mega-SEC method provides an efficient and powerful new approach for the biochemical analysis of mammalian polysomes and ribosome subunits, isolated from extracts of either cultured cell lines, or tissues. Furthermore, we note that this approach using large pore size (2,000 Å) SEC columns and uHPLC chromatography, potentially has many additional applications beyond the field of protein translation. For example, the ability to

efficiently and rapidly separate very large structures by Mega-SEC could also be applied to the purification and biochemical analysis of other large sub-cellular complexes, viral particles and secreted vesicles such as exosomes. A particular advantage of using SEC is that the complexes are isolated rapidly and in buffers suitable for performing biochemical assays immediately on the isolated material without the need for further purification.

# Materials and methods

## Key resources table

| Reagent type (species) or resource | Designation | Source or reference | Identifiers | Additional information |
|---|---|---|---|---|
| Cell line (human) | HeLa | ATCC | RRID:CVCL_0030 | Tested negative for mycoplasma |
| Cell line (human) | U2OS | ATCC | RRID:CVCL_0042 | Tested negative for mycoplasma |
| Cell line (human) | HCT116 p53 +/+ | ATCC | RRID:CVCL_0291 | Tested negative for mycoplasma |
| Cell line (human) | HCT116 p53 -/- | Horizon | RRID:CVCL_S744 | Tested negative for mycoplasma |
| Biological sample (*Mus musculus*) | C57BL/6J | Australian BioResources | RRID:IMSR_JAX:000664 | |
| Antibody | Anti-Ribosomal protein S10 antibody [EPR8545] | Abcam Cat# ab151550 | RRID:AB_2714147 | |
| Antibody | Ribosomal Protein L28 (A-16) antibody | Santa Cruz Biotechnology Cat# sc-14151 | RRID:AB_2181749 | |
| Antibody | RPL14/Ribosomal Protein L14 Antibody | Bethyl Cat# A305-052A | RRID:AB_2621246 | |
| Antibody | RPLP0 antibody | Abcam Cat# ab88872 | RRID:AB_2042838 | |
| Antibody | Anti-mouse IgG, HRP-linked Antibody | Cell Signaling Technology Cat# 7076 | RRID:AB_330924 | |
| Antibody | Anti-rabbit IgG, HRP-linked Antibody | Cell Signaling Technology Cat# 7074 | RRID:AB_2099233 | |
| Antibody | Anti-Goat IgG (whole molecule) -Peroxidase antibody produced in rabbit | Sigma-Aldrich Cat# A5420 | RRID:AB_258242 | |

## Materials

BioBasic SEC-300, SEC-1,000 and SEC-2,000 LC Columns were from Thermo Fisher Scientific (We note that the SEC-2,000 LC Column was not commercially available when these experiments were carried out). Agilent Bio SEC-5 2,000 Å was from Agilent. CHAPS (Sol-Grade) was from Anatrace. DMEM (high glucose, pyruvate; 41966–029), Fetal bovine serum (FBS), dialyzed FBS, EBSS (with calcium, magnesium and phenol red), MEM amino acids solution (50X), MEM non-essential amino acids solution (100X), MEM vitamin solution (100X), L-glutamine (200 mM), Penicillin-Streptomycin, SUPERase In RNase inhibitor, TRIzolLS reagent, BCA protein assay kit, EZQ protein quantitation kit, CBQCA protein quantitation kit, and Chemiluminescent Nucleic Acid Detection Module kit were from Thermo Fisher Scientific. Complete EDTA-free Protease inhibitor was from Roche. 0.45 µm Ultrafree-MC HV centrifugal filter units and Stericup filter units were from Merck Millipore. PVDF membrane and Hybond-N +nylon membrane were from GE Healthcare Life Sciences. IMMUNO SHOT was from Cosmo Bio. Biotin-dC-Puromycin was from Jena Bioscience. Any other standard laboratory chemicals were obtained from either Sigma, or VWR.

## Cell culture

HeLa (RRID: CVCL_0030), HCT116 p53+/+ (RRID: CVCL_0291), and U2OS (RRID: CVCL_0042) cells were purchased from ATCC. HCT116 p53-/- cells (RRID: CVCL_S744) was purchased from Horizon. Cell lines were tested negative for mycoplasma contamination.

Cells were maintained in DMEM supplemented with 10% FBS, penicillin (100 U/ml) and streptomycin (0.1 mg/ml) at 37°C and 5% $CO_2$. For the starvation experiments, HeLa cells were washed twice with PBS and incubated for 2 hr in EBSS supplemented with 10% dialyzed FBS, vitamins,

glucose (4.5 g/L) and sodium pyruvate, either with, or without, a mixture of amino acids (MEM amino acids solution, NEAA, and glutamine), at almost the same concentration of DMEM. Cells were treated with 50 µg/ml cycloheximide to maintain polysome stability for 5 min under 37°C and 5% $CO_2$ before harvest.

## Mice

C57BL/6J male mice were purchased from Australian BioResources (Moss Vale, Australia) at 7 weeks of age and kept at 22°C on a 12 hr light/dark cycle with free access to food (CHOW - 13% calories from fat, 22% calories from protein, and 65% calories from carbohydrate, 3.1 kcal/g; Gordon's Specialty Stock Feeds, Yanderra, Australia) and water. Mice were sacrificed in the fed state (9am) at 10–12 weeks of age by cervical dislocation, and the liver tissues were quickly excised and freeze-clamped in liquid nitrogen. The tissues were stored at −80°C before analysis. All experiments were carried out with the approval of the University of Sydney Animal Ethics Committee (2016/1096), following guidelines issued by the National Health and Medical Research Council of Australia.

## Preparation of cell/tissue lysates for SEC

Cells in two 10 cm dishes (80% confluency) were grown as described above, washed with ice-cold PBS containing 50 µg/ml cycloheximide, scraped on ice, collected by centrifugation, lysed by vortexing for 10 s in 300 µl of polysome extraction buffer (20 mM Hepes-NaOH (pH 7.4), 130 mM NaCl, 10 mM $MgCl_2$, 1% CHAPS, 0.2 mg/ml heparin, 2.5 mM DTT, 50 µg/ml cycloheximide, 20 U SUPERase In RNase inhibitor, cOmplete EDTA-free Protease inhibitor) or in polysome extraction buffer containing 0.5 M NaCl, incubated for 15 min on ice, and centrifuged at 17,000 $g$ for 10 min (all centrifugations at 4°C). Frozen liver (typically 100 mg wet weight) was homogenized in 2 ml of polysome extraction buffer using a glass dounce tissue grinder with 10 strokes. After clarification by 2,500 $g$ for 10 min followed by removal of the lipid layer on the top of the supernatants, the lysates were mixed with heparin (1 mg/ml) and centrifuged at 16,000 $g$ for 10 min. Supernatants from either cells, or liver tissue, were filtered through 0.45 µm Ultrafree-MC HV centrifugal filter units by 12,000 $g$ for 2 min, and protein and RNA amounts in the filtrates were quantified by BCA protein assays and Bio-Photometer (Eppendorf), respectively. To dissociate ribosomes, the filtrates were treated with 30 mM EDTA and filtered again before injecting to SEC.

## SEC

Columns employed were all of dimensions 7.8 × 300 mm with 5 µm particles and included Thermo BioBasic SEC 300 Å, 1,000 Å, or 2,000 Å columns, or Agilent Bio SEC-5 2,000 Å columns. Using a Dionex Ultimate 3,000 Bio-RS uHPLC system (Thermo Fisher Scientific), each SEC column was equilibrated with two column volumes (CV) of filtered SEC buffer (20 mM Hepes-NaOH (pH 7.4), 60 mM NaCl, 10 mM $MgCl_2$, 0.3% CHAPS, 0.2 mg/ml heparin, 2.5 mM DTT) (all column conditioning and separation at 5°C) and 100 µl of 10 mg/ ml of filtered bovine serum albumin (BSA) solution diluted by PBS was injected once to block the sites for non-specific interactions. After monitoring the column condition by injecting standards, including 10 µl of 10 mg/ml BSA solution and 25 µl of Hyper-Ladder 1 kb (BIOLINE), cell lysates were injected onto a column. For the separation of EDTA-treated cell lysates, SEC buffer containing 30 mM EDTA instead of 10 mM $MgCl_2$ was used. For the separation of the high salt-extracted lysates, SEC buffer containing 0.5 M NaCl instead of 60 mM NaCl was used. For the comparison of the polysome profile from the cells treated either with, or without, EDTA or amino acids, 150 µl of cell lysate containing 50 µg of RNA from each sample was injected. For the separation of polysomes from liver tissue, 180 µl of tissue lysate containing 150 µg of RNA was injected for the check of reproducibility and 200 µl of tissue lysate containing 2 mg of protein was injected and separated with SEC buffer not containing heparin for the proteomic analysis. The chromatogram was monitored by measuring UV absorbance at 215, 260 and 280 nm with 1 Hz of data collection rate by the Diode Array Detector. The flow rate was 0.8 ml/min and either 48 × 100 µl fractions, 24 × 200 µl fractions or 16 × 300 µl fractions were collected from 9 min to 14.6 min using a low-protein binding 96-deep-well plate 1 mL (Eppendorf) at 4°C. The peaks were quantified using the Chromeleon 6.8 Chromatography Data System (Thermo Fisher Scientific). Proteins in each fraction were precipitated with 10% trichloroacetic acid (TCA) and 10 µg of BSA for western blotting. Preparation for the proteomic analysis was described below. RNAs in each fraction were extracted

either by TRIzol LS reagent, or by treating the samples with 0.1 mg/ml proteinase K, 1% SDS and 15 mM EDTA for 1 hr at 42°C followed by phenol-chloroform extraction, and precipitated with isopropanol containing 5 µl of 0.25% linear acrylamide.

For the column cleanup, after flushing water for at least 10 CV at room temperature, the column was first filled with the pepsin solution (20 µg/ml pepsin, 0.1 M acetic acid, 0.5 M NaCl) by running for at least 6 CV and incubated for 1 hr at 37°C without flow. Next, the RNase A solution (20 µg/ml RNase A, 10 mM Tris-HCl (pH 7.4), 0.5 M NaCl) was filled as above and the column was incubated for 1 hr at 37°C without flow. Then, the column was flushed by 1% SDS, 2 x SSC (0.3 M NaCl, 30 mM trisodium citrate dehydrate) solution for 10 CV at 50°C. After rinsing with 10 CV dH$_2$O at room temperature, the column was equilibrated with the SEC buffer.

## Protein digestion, peptide clean-up and LC-MS/MS analysis

Proteins in each fraction were reduced and alkylated with TCEP and NEM then were cleaned up using the SP3 method (*Hughes et al., 2014*) and 'on-beads' digestion to peptides performed using trypsin, added at a ratio of 1:50 by weight, based upon an EZQ protein assay of the fractions, then incubated for 18 hr at 37°C. Peptides were then recovered by SP3 and further purified with SDB-RPS Stagetips (*Rappsilber et al., 2007*). SDB-RPS StageTips were generated by punching double-stacked SDB-RPS discs (Sigma, Cat#66886 U) with an 18-gauge needle and mounted in 200 µl tips (Eppendorf). Each tip was wetted with 100 µL of 100% acetonitrile and centrifuged at 1,000 x g for 1 min. Following wetting, each StageTip was equilibrated with 100 µL of 30% methanol/1% TFA and 0.1% TFA in H2O, with centrifugation for each at 1,000 x g for 3 min. Each StageTip was then loaded with the equivalent of ~5 µg peptide in 1% TFA and centrifuged as above. The peptides were washed once with 100 µl of 0.2% TFA in water, which was followed by one wash with 100 µl of 99% isopropanol/1% TFA with and centrifuged after each step as above. To elute, 100 µL of 5% ammonium hydroxide/80% acetonitrile was added to each tip and centrifuged as above for 5 min. Eluted peptides were dried using a GeneVac EZ-2 using the ammonia setting at 35°C for 1 hr. Dried peptides were resuspended in 20 µL of 5% formic acid and stored at 4°C until analysed by LC-MS. Peptide concentrations were determined using a CBQCA assay.

Peptides were analyzed by a Dionex RSLCnano HPLC-coupled Q-Exactive Plus mass spectrometer (Thermo Fisher Scientific). Using a Thermo Fisher Dionex RSLCnano UHPLC, peptides in 5% (vol/vol) formic acid (injection volume 3 µL) were directly injected onto a 45 cm x 75 um C18 (Dr. Maisch, Ammerbuch, Germany, 1.9 µm) fused silica analytical column with a ~10 µm pulled tip, coupled online to a nanospray ESI source. Peptides were resolved over gradient from 5% acetonitrile to 40% acetonitrile over 60 min with a flow rate of 300 nL min−1. Peptides were ionized by electrospray ionization at 2.3 kV. Tandem mass spectrometry analysis was carried out on a Q-Exactive Plus mass spectrometer using HCD fragmentation. The data-dependent acquisition method used acquired MS/MS spectra on the top 30 most abundant ions at any one point during the gradient. Briefly, the primary mass spectrometry scan (MS1) was performed in the Orbitrap at 70,000 resolution. Then, the top 30 most abundant m/z signals were chosen from the MS1 for collision-induced dissociation in the HCD cell and MS2 analysis in the Orbitrap at 17,500 resolution. Precursor ion charge state screening was enabled and all unassigned charge state, or single charge, were rejected.

## MS data analysis

The RAW data produced by the mass spectrometer were analyzed using the MaxQuant quantitative software package (Version 1.5.1.3) (*Cox and Mann, 2008*). This version of MaxQuant includes an integrated search engine, Andromeda (*Cox et al., 2011*). Peptide and Protein level identification were both set to a false discovery rate of 1% using a target-decoy based strategy. The database supplied to the search engine for peptide identifications was the Mouse Swissprot database. The mass tolerance was set to 4.5 ppm for precursor ions and MS/MS mass tolerance was set at 20 ppm. Enzyme was set to trypsin (cleavage C-terminal to lysine and arginine) with up to two missed cleavages. Deamidation of Asn and Gln, oxidation of Met, pyro-Glu (with peptide N-term Gln), protein N-terminal acetylation were set as variable modifications. N-ethylmaleimide on Cys was searched as a fixed modification. The output from MaxQuant provided peptide level data as well as protein group level data. We used the protein groups as defined by the Maxquant package. iBAQ (Intensity Based Absolute Quantification) algorithm in MaxQuant was used for protein quantitation. All the

RAW MS data have been deposited to the ProteomeXchange Consortium via the PRIDE partner repository with the data set identifier PXD008913. The MaxQuant output has also been uploaded to the ProteomeXchange Consortium under the same identifier given above.

MaxQuant output was analyzed in RStudio v1.0.136 with R language (version 3.3.2). The iBAQ intensity profile for each replicate was smoothed using a three-fraction sliding mean and the minima and maxima of each profile was normalized within the limits 0 and 1, respectively. The mean and range for each protein, across two biological replicates, was calculated for subsequent plotting using the ggplot2 package (http://ggplot2.org/), correlation analysis, and basic clustering. Proteins labeled as either contaminants, or reverse hits, were removed from the analysis. The mean profiles for each protein were hierarchically clustered. The basic hierarchical clustering was performed using the Euclidean distance measurement and a 'complete' agglomeration method. The tree calculated for each data set was cut to generate clusters with a mean Pearson correlation coefficient of 0.95.

## Western blotting

Proteins were separated by SDS-PAGE and electrophoretically transferred to a PVDF membrane. The membrane was blocked with 3% non-fat dried skim milk in TBS containing 0.1% (w/v) Tween 20 (TBST) for 1 hr at room temperature, washed twice with TBST for 5 min and incubated with the appropriate primary antibody, using IMMUNO SHOT, at 4°C overnight. After washing three times 10 min with TBST, the membranes were incubated with a secondary antibody, conjugated with horseradish peroxidase (HRP), using IMMUNO SHOT for 1 hr at room temperature and then washed a further three times 10 min with TBST. Chemiluminescent-stained proteins were detected by LAS4000 image analyzer (Fujifilm).

## Antibodies

The antibody sources and dilution ratios used for western blotting in this study were as follows: rabbit polyclonal anti-RPS10 (Abcam, ab151550; 1:3,000), goat polyclonal anti-RPL28 (A-16) (Santa Cruz Biotechnology, sc-14151; 1:500), rabbit polyclonal anti-RPL14 (Bethyl, A305-052A; 1:5,000), mouse polyclonal anti-RPLP0 antibody (Abcam, ab88872; 1:2,000), HRP-conjugated anti-mouse IgG (Cell Signaling Technology, #7076; 1:10,000), HRP-conjugated anti-rabbit IgG (Cell Signaling Technology, #7074; 1:10,000), HRP-conjugated anti-goat IgG (Sigma, A5420; 1:10,000).

## *In vitro* puromycylation

*In vitro* puromycylation was performed as previously reported with a slight modification (*Aviner et al., 2013*; *Aviner et al., 2014*). Each fraction separated by SEC was equally divided into two and 100 pmol of puromycin was added to the one and 100 pmol of Biotin-dC-puromycin was added to the other. After incubation for 15 min at 37°C, the samples were precipitated by TCA, separated by SDS-PAGE and transferred to a PVDF membrane. The membrane was blocked with 3% non-fat dried skimmed milk in TBST for 1 hr at room temperature, washed with TBST for 10 min twice, and incubated with HRP-conjugated streptavidin in TBST for 15 min at room temperature. After washing 3 × 10 min with TBST, biotin-puromycin conjugated nascent polypeptide chains were detected using a LAS4000 image analyzer.

## Northern blotting

Northern blotting was performed as previously reported with a slight modification (*Yoshikawa et al., 2015*). Briefly, RNAs were electrophoresed on a 1% agarose gel including 1.2% formaldehyde and 1 x TT buffer (*Mansour and Pestov, 2013*; *Preti et al., 2013*) and transferred to a Hybond-N+ nylon membrane, which was subsequently dried and then UV cross-linked. After staining with methylene blue, RNAs were hybridized, either to biotin-labeled DNA oligonucleotide probes at 50°C, or to biotin-labeled oligo dT probe (iba) at 42°C, overnight in PerfectHyb Plus hybridization buffer (Sigma). The hybridized membrane was washed sequentially with 2 x SSC containing 0.1% SDS for 5 min at room temperature, 0.5 x SSC containing 0.1% SDS for 20 min at 50°C or 42°C and 0.1 x SSC containing 0.1% SDS for 20 min at room temperature. The hybridized RNA was detected using a Chemiluminescent Nucleic Acid Detection Module kit and a LAS4000 image analyzer. The sequences of oligonucleotides used as probe were follows: for *ACTB* mRNA, 5'-CTCC TTAATGTCACGCACGAT-3'; for *RPL23a* mRNA, 5'- GTTGACCTTGGCCACATCAATGTC-3'; for 18S

rRNA, 5'- GGCGACTACCATCGAAAGTTGATAG −3'; for 28S rRNA, 5'- TTCGGAGGGAACCAGC TACTAGAT −3'. 3'-end biotin-labeled oligonucleotides were synthesized by Eurofins Genomics.

## Sucrose density gradient (SDG) analysis

SDG analysis was performed as previously reported with a slight modification (*Ishikawa et al., 2017*; *Strezoska et al., 2000*). Briefly, the supernatants containing 100 or 200 µg RNAs in 500 µl, prepared as above, were layered on 10 ml of 10%–45% (wt/wt) SDGs in 20 mM Hepes-NaOH (pH 7.4), 60 mM NaCl, 10 mM $MgCl_2$, 1 mM DTT and 0.1 mg/ml heparin, which were prepared by a simple-diffusion-based method (*Pestov et al., 2008*), then centrifuged at 36,000 rpm for 3 hr at 4°C in a Beckman SW41Ti rotor. The samples were fractionated into 21 fractions (each ~500 µl) using Density Gradient Fractionation Systems (TELEDYNE ISCO), with continuous measurement of the absorbance at 254 nm. Proteins and RNAs in each fraction were extracted as described above.

## Electron microscopy analysis

Samples were negatively stained with 2% uranyl acetate. In a typical experiment, a freshly glow discharged carbon coated grid was first soaked on a 5 µl drop of respective sample for 3 min, excess solvent removed and quickly stained with uranyl acetate for 1 min. Subsequently, the grid was washed four times with a 5 µl water drop to remove any excessive stain and air dried. The grid was then mounted on a JEOL side entry room temperature probe and loaded on to the electron microscope for subsequent viewing and imaging, using a JEOL 2010F with FEG (Field Emission Gun) electron microscope operating at 200kV. The imaging was carried out in TEM mode at a nominal magnification of 40,000X and the micrographs were recorded on a 4K × 4K Gatan CCD camera.

## Statistical analysis

To analyze reproducibility between replicates, either the UV absorbance at 260 nm between the times indicated for each replicate for SEC profiles, or the UV absorbance at 254 nm between the times indicated for each replicate for SDG profiles, was correlated between replicates. Image Studio Lite Ver 5.2 (LICOR) was used to quantify the band intensity. The standard deviation for the three biological replicates was calculated and used to provide either standard error bars on the graphs, or error ribbons on the UV chromatograms. An unpaired *t*-test was used to decide significant differences.

## Acknowledgements

We thank Thermo Fisher Scientific for access to the prototype SEC-2000 LC Columns. We thank the Cowling Lab (University of Dundee) for the use of the SDG fraction collection device. We thank SydneyMS for the provision of mass spectrometry instrumentation used in this study. We thank J Hukelmann and other colleagues in the Lamond laboratory for valuable suggestions. HY has been funded by the Naito Foundation Grant for Overseas, the Uehara Memorial Foundation Postdoctoral Fellowship and is currently supported by the European Union's Horizon 2020 research and innovation programme under the Marie Sklodowska-Curie Individual Fellowship. ML was funded by the Royal Society of Edinburgh and Scottish Government Personal Research Fellowship and is currently supported by the Cancer Institute NSW Future Research Leader Fellowship. TL is currently supported by the Wellcome Trust and the Royal Society Sir Henry Dale Fellowship. This work is supported by Wellcome Trust grants (073980/Z/03/Z; 105024/Z/14/Z and 098503/E/12/Z), a Wellcome Trust strategic award (097945/B/11/Z), and a National Health and Medical Research Council of Australia (NHMRC) Project Grant GNT1120475.

## Additional information

### Funding

| Funder | Grant reference number | Author |
|---|---|---|
| H2020 Marie Skłodowska-Curie Actions | Individual Fellowship 657087 | Harunori Yoshikawa |

| Uehara Memorial Foundation | Postdoctoral Fellowship 201430061 | Harunori Yoshikawa |
| Naito Foundation | Grant for Overseas 2013-413 | Harunori Yoshikawa |
| Royal Society of Edinburgh | Personal Research Fellowship | Mark Larance |
| Cancer Institute NSW | Future Research Leader fellowship | Mark Larance |
| National Health and Medical Research Council | Project Grant GNT1120475 | Mark Larance |
| Scottish Government | Personal Research Fellowship | Mark Larance |
| Wellcome | Sir Henry Dale Fellowship 206211/Z/17/Z | Tony Ly |
| Wellcome | Strategic Award 097945/B/11/Z | Tom Owen-Hughes Angus I Lamond |
| Wellcome | 073980/Z/03/Z | Angus I Lamond |
| Wellcome | 105024/Z/14/Z | Angus I Lamond |
| Wellcome | 098503/E/12/Z | Angus I Lamond |

The funders had no role in study design, data collection and interpretation, or the decision to submit the work for publication and therefore the contents of the published material are solely the responsibility of the individual authors and do not reflect the views of the funders.

## Author contributions

Harunori Yoshikawa, Conceptualization, Data curation, Funding acquisition, Investigation, Methodology, Writing—original draft, Writing—review and editing; Mark Larance, Conceptualization, Data curation, Investigation, Visualization, Funding acquisition, Methodology, Writing—original draft, Writing—review and editing; Dylan J Harney, Data curation; Ramasubramanian Sundaramoorthy, Data curation, Writing—review and editing; Tony Ly, Conceptualization, Methodology, Writing—review and editing; Tom Owen-Hughes, Funding acquisition, Writing—review and editing; Angus I Lamond, Conceptualization, Supervision, Funding acquisition, Project administration, Writing—review and editing

## Author ORCIDs

Harunori Yoshikawa https://orcid.org/0000-0003-3793-6219
Tony Ly http://orcid.org/0000-0002-8650-5215
Tom Owen-Hughes http://orcid.org/0000-0002-0618-8185
Angus I Lamond http://orcid.org/0000-0001-6204-6045

## Ethics

Animal experimentation: All experiments were carried out with the approval of the University of Sydney Animal Ethics Committee (2016/1096), following guidelines issued by the National Health and Medical Research Council of Australia.

## Decision letter and Author response

Decision letter https://doi.org/10.7554/eLife.36530.028
Author response https://doi.org/10.7554/eLife.36530.029

# Additional files

### Supplementary files

• Supplementary file 1. Protein level data identified in mouse liver tissue, classified by cluster. The table summarizes the proteins identified in mouse liver tissue and includes the following data for

each protein identification: protein ID, protein name, Gene name, cluster number, total iBAQ intensities from two biological replicates and individual intensities.

DOI: https://doi.org/10.7554/eLife.36530.023

• Transparent reporting form

DOI: https://doi.org/10.7554/eLife.36530.024

### Data availability

All of the RAW MS data and MaxQuant-generated output have been deposited to the ProteomeXchange Consortium via the PRIDE partner repository with the dataset identifier PXD008913.

The following dataset was generated:

| Author(s) | Year | Dataset title | Dataset URL | Database, license, and accessibility information |
|---|---|---|---|---|
| Larance M, Harney D, Yoshikawa H, Ly T, Lamond AI | 2018 | Ribo Mega-SEC Provides a Rapid, Efficient and Reproducible Analysis of Mammalian Polysomes and Ribosomal Subunits | https://www.ebi.ac.uk/pride/archive/projects/PXD008913 | Publicly available at EBI PRIDE (accession no. PXD008913) |

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
