## [Decision Letter]

[Editors’ note: a previous version of this study was rejected after peer review, but the authors submitted for reconsideration. The first decision letter after peer review is shown below.]

Thank you for submitting your work entitled "Ribo Mega-SEC Provides Efficient and Reproducible Analysis of Polysomes and Ribosomal Subunits" for consideration by *eLife*. Your article has been reviewed by three peer reviewers, including a member of our Board of Reviewing Editors, and the evaluation has been overseen by a Senior Editor. The following individual involved in review of your submission has agreed to reveal their identity: Leos Shivaya Valasek (Reviewer #3).

Our decision has been reached after consultation between the reviewers. Based on these discussions and the individual reviews below, we regret to inform you that your work cannot be considered further for publication in *eLife*.

Summary of study:

This study describes the use of HPLC size exclusion chromatography to resolve polysomes from monosomes and free subunits in whole cell extracts. The technique (referred to here as RM-SEC) appears to have several advantages over sucrose density gradient centrifugation (SDGC), such as greater rapidity and (possibly) reproducibility, and fractionation in physiological buffers versus sucrose. They present evidence that the technique gives reproducible separations consistent with SDGC, with the exception that the 80S (monosome) peak is dramatically reduced in this technique. They favor the idea that the 80S peak is overemphasized in SDGC at least partly due to artefactual dissociation of polysomes into 80S monosomes during centrifugation.

Summary of critique: All three reviewers agreed RM-SEC could be a valuable addition to translation research, at least for its rapidity and amenability to high-throughput applications, and for the absence of high concentrations of sucrose for downstream cryo-EM applications. However, all three also voiced serious concerns with the quality of the data and interpretations of certain results. It was agreed that the inability to resolve different n-mer polysomes from one another is an important drawback of the technique that the authors should attempt to resolve. The very minimal 80S peak is also troubling and it was felt that further work is needed to illuminate the dramatic difference between RM-SEC and SDGC in terms of 80S abundance, as this will be crucial to the acceptance of RM-SEC by the translation field. In this regard, it was felt that you should attempt to demonstrate that RM-SEC will detect an extensive conversion of polysomes to inititation-arrested 80S monosomes engendered by Harringtonin, and to compare the high-salt sensitivity of the 80S peaks in SDGC vs RM-SEC. It is also important to present evidence supporting your assertion that a substantial fraction of polysomes dissociate into 80S monosomes during SDGC, as this calls into question a large body of work in the field based on SDGC, starting with the presentation of your own data from an experiment in which polysomes isolated by RM-SEC were resolved by SDGC. In addition, the SDGC separation in Figure 1—figure supplement 2, used to assign ribosomal species in RM-SEC is of poor quality, which seemed likely to have overemphasized contamination of the 80S peak with free 60S subunits using SDGC, and would have to be improved. Another drawback is working with cells in which the polysome content is very low, and it was felt that you should examine rapidly growing yeast extracts to remedy this problem-especially considering the importance of yeast as a model system for elucidating translation mechanisms.

There are also numerous issues with inexplicable results from the analysis of rRNA and ribosomal proteins across the RM-SEC fractions, including: (i) in Figure 1B-EDTA-treated lysate, fraction 8 (40S peak) contains a considerable amount of 18S rRNA but no RPS10. (ii) In Figure 1—figure supplement 5-EDTA-treated lysate, fraction 3 (40S) contains no 28S rRNA but a lot of RPLP0. (iii) In Figure 1—figure supplement 5, there is no peak of Rps10 or 18S rRNA in the 80S peak in fraction 6. And in Figure 3C, the redistribution of ACTB mRNA from polysomes to monosomes appears to be nearly as dramatic as that seen for uL23 mRNA, necessitating quantification of these mRNA distributions from replicates. The discrepancy between the results in Figure 4 and those published by Gao et al., 2015 could be explored further by comparing in your their own hands the results obtained using SDGC and RM-SEC; although this would not be considered essential for acceptance. Finally, it is necessary to substantiate your claim that the results of SDGC are less reproducible than from RM-SEC.

In view of the many criticisms and requests for additional experiments, the paper must be rejected at this stage. However, in view of the strong potential of this technique for improving the analysis and quantification of polysome distributions, all three reviewers would be willing to serve as referees again to evaluate a new submission containing considerable new experimental work that addresses all of the major issues noted above.

Reviewer #1:

This study describes the use of HPLC size exclusion chromatography to resolve polysomes from monosomes and free subunits. It appears to have several advantages over sucrose density gradient centrifugation (SDGC), such as greater rapidity and reproducibility, and fractionation in physiological buffers versus sucrose. They present evidence that the technique gives reproducible separations consistent with SDGC, with the exception that the 80S (monosome) peak is dramatically reduced in this technique. They favor the idea that the 80S peak is overemphasized in SDGC at least partly due to artefactual dissociation of polysomes into 80S monosomes during centrifugation.

This appears to be a very significant technological advance for the translation field. However, questions surrounding the 80S monosome peak could hinder workers from embracing it as an alternative to SDGC, since determining polysome:monosome ratios as a measure of translational activity is frequently a goal of these experiments. As such, the authors should be asked to substantiate their interpretation by presenting the data cited but not shown for the experiment they describe in the Results and Discussion in which the polysome fraction isolated from HPLC was resolved by SDGC. According to their explanation, the HPLC-polysome fraction should yield a substantial proportion of 80S monosomes when separated in SDGC.

In view of recent findings from Heyer and Moore that a large fraction of the monosomes are involved in translation elongation, it seems possible that the vacant 80S couples unbound to mRNA thought to constitute a significant fraction of the 80S peak in SDGC dissociate into free subunits in the HPLC separation. If so, essentially all of the small 80S peak resolved by HPLC would be comprised of translating 80S ribosomes and hence, resistant to high salt; whereas the a substantial proportion of the 80S peak in SDGC is known to dissociate in high-salt owing to the presence of vacant 80S couples. It seems incumbent on the authors to examine this question by comparing the amount of monosomes present in HPLC versus SDGC separations in high salt conditions where vacant couples dissociate into free subunits.

Reviewer #2:

The manuscript by the Lammond lab on Ribo mega-SEC describes a quick method to separate polysomal complexes, which can be used as an alternative to the more laborious and classical sucrose gradient method. The SEC method is robust and reproducible, and has the potential to become a benchmark for any lab working on translational control. There are some details, however, that need to be clarified:

1) It is striking that the 80S peak is almost invisible in most SEC profiles compared to sucrose gradients. Further, in Figure 1—figure supplement 2 each sucrose gradient peak/fraction from U2OS cells is resolved by SEC, and considerable cross-contamination of 60S and 80S subunits is found. For example, the 80S peak in sucrose gradients is found to contain more 60S than 80S by SEC. This is important. Researchers working on translational control and ribosome biogenesis have made conclusions based on the identity of peaks in sucrose gradients, an identity that has been supported by the use of translation inhibitors. Are these conclusions now wrong? For this reason, I think it is relevant to ascertain the composition of the peaks by other methods (e.g. throw the peak material onto an electron microscopy grid to visualize 60S versus 80S ribosomes or mixed assemblies).

2) Regarding the issue above, how would profiles in the presence of Harringtonine (which should increase the 80S peak) look like in sucrose gradients compared with SEC? In these conditions, how does the 80S peak from the sucrose gradient appear in SEC? The experiment could be performed using Hela cells for a direct comparison with Figure 1A.

3) The rRNA profiles do not always coincide with the RP profiles. For example, in Figure 1B-EDTA-treated lysate, fraction 8 (40S peak) contains a considerable amount of 18S rRNA but no RPS10. Similarly, in Figure 1—figure supplement 5-EDTA-treated lysate, fraction 3 (40S) contains no 28S rRNA but a lot of RPLP0.

Reviewer #3:

In the presented paper, Yoshikawa et al. present Ribo Mega-SEC, a newly developed approach for the separation and biochemical analysis of polysomes and ribosomal subunits using size Exclusion Chromatography and uHPLC. Based on the presented data I agree that it is an interesting alternative technique to the classical SDG, however, I disagree that this method represents a significant improvement of the polysomal analysis in general, mainly with respect to the efficiency, reproducibility and downstream applications.

Major comments:

– Resolution of polysomes by Mega-SEC is very poor – they come in one broad peak, in principle also including the monosome, when compared to beautifully separated n-somes by the conventional SDG.

– Figure 1—figure supplement 2B; the second (80S) and third (60S) peaks are not resolved well enough; the 80S peak contains a very large amount of 60S subunits and vice versa.

– I also disagree with the optimal flow rate and urge the authors to test also a lot slower rates (even 0.5 ml/min is not satisfactory at all); extending the analysis from 15 min to 30 min would still preserve the time-spent advantage but at the same it could dramatically improve the resolution that is a real problem with this technique.

– According to my own (rather extensive) experience, the amount of polysomes obtained by SDG by these authors is exceptionally low (see Figure 1–figure supplement 2A and 5). It almost looks like a very efficient polysomal run-off, why is that? In Figure 1—figure supplement 5 there is also a surprisingly small peak of 18S and RPS10 in the 80S-containing fractions.

– I also disagree that the conventional SDG is troublesome with respect to reproducibility and its usage for downstream applications.

– Figure 3C; I see even more dramatic difference in the ACTB distribution here – both fat peaks are gone?

– “[…] note that an alternative study, using SDG, reported that polysomes are disassembled in fasted mouse liver (Gao et al., 2015)." I would expect this to be true but one never knows. Nonetheless, this discrepancy may be an important one, worth pursuing further (examining this phenomenon by both techniques in parallel), as its resolution may bring a real advantage of Mega-SEC over SDG into life.

In summary, I am not sure what advantage (apart for rough high-throughput screening) this technique may bring to the people planning to analyse polysomes over the SDG technique. For example, the authors should, after the very problematic "resolution issue" has been resolved, analyze the distribution of as many translation factors (eIFs, eEFs and eRFs) as possible across the entire profile under various conditions and show that their distribution changes and that these changes are highly reproducible. This would be a real improvement that would most probably change for example my mind to give the Mega-SEC a try.

[Editors’ note: what now follows is the decision letter after the authors submitted for further consideration.]

Thank you for resubmitting your work entitled "Efficient Analysis of Mammalian Polysomes in cells and tissues using Ribo Mega-SEC" for further consideration at *eLife*. Your revised article has been favorably evaluated by James Manley (Senior Editor), and 3 reviewers, including a member of our Board of Reviewing Editors.

The reviewers appreciated the large amount of effort that was expended to address the concerns that were raised with the previous version of the paper, and agree that the paper was considerably strengthened by these efforts. There are, however some remaining issues. First, although the results shown in Figures 1A and C provide strong evidence that the SEC technique is capable of resolving polysomes of different sizes, none of the separations presented in the paper actually show a resolution of the single polysome peak into discrete n-mer peaks in the manner that is routinely achieved by SDGC. Identification of the distribution of the different n-mer polysomes by SDGC provides valuable information about the rate of translation initiation and also is crucial in experiments that seek to identify the number of ribosomes per mRNA on particular mRNAs by determining the distribution of individual mRNAs across the different n-mer species, either by RT-PCR or Northern analysis of particular mRNAs or by microarray or RNA-seq analysis across the fractions. It is felt therefore that if the authors can provide in the results an SEC separation in which different n-mer species can be resolved, even if this requires a significantly longer separation time than achieved with the 0.8ml/min flow rate chosen for most experiments, this would greatly increase the likelihood that this new approach would be widely embraced by the translation field and that this paper would be highly cited in the future. In addition to this experiment, there are a number of other requests for additional data, explanation of certain results, or corrections/clarifications to the text raised by each of the three reviewers that must be addressed in a revised manuscript, as described in the individual reviews.

Reviewer #1:

I am satisfied that the addition of new data revisions of text and figures have adequately addressed all of the major concerns with the previous version. There are however some errors in citing figures or the relevant panels of figures that need to be corrected as follows:

– Results section, paragraph three: cite specific panels of Figure 1 and Figure 1—figure supplement 2 rather than the entire figures to refer to specific results.

– Results section, paragraph ten.: Again, cite panels A and B of Figure 3—figure supplement 1 where appropriate rather than just citing the whole figure. Check entire manuscript for proper citations of specific panels of figures throughout.

– Results section, paragraph eighteen: shouldn't this cite Figure 6—figure supplement 3A vs 3B?

Reviewer #2:

The manuscript has significantly improved. The authors carried out a lot of new valuable experiments that have convinced me this technique can be used as an alternative to the classical SDGC in some – but not all – applications. Nonetheless, I still have many concerns – outlined below – that should be addressed before this paper introducing a novel technique can be accepted for publication.

First my personal point of view as a researcher for whom the SDGC represents a daily bread – this is not a critique, just a practical guide into the "supply and demand" issue.

"Speed is not all what matters."

Truly, I do not see a practical use of Mega-SEC in most of our applications. We work with human as well as yeast cells and what we need is the following.

1) Prepare polysome profiles with nicely resolved polysomes to calculate the P/M ratio and obtain an insight into the fitness of translation as a whole, which the new technique is not able to offer.

2) To study various aspects of the initiation process, we need to clearly resolve the 43-48S pre-initiation complexes (PICs) from the top of the gradient where free factors sediment, as well as from heavier 60S species, to be able to collect 10-12 fractions that we subject to Western blotting to quantitatively analyze the distribution of various initiation factors across the gradient in wt vs. mutant cells (employing Mass-Spec is an interesting idea but it could not be used to quantify changes in the distribution of factors). To achieve this we have to use the 7.5-30% linear sucrose gradient instead of 5-45%. The separation of the 40S species from the lighter material by Mega-SEC does not seem to be sufficient to manage this but perhaps it could if the flow rate was a lot slower?

3) To study various aspects of the termination process, we need to clearly separate heavy from light polysomes and Mega-SEC does not allow any separation at all (they all come as a single peak in 3 fractions). I understand that they are all there and they are stable, as demonstrated, but it does not help.

4) I agree that it could be perhaps used for ribosome profiling but to set-up a brand new method in the lab just because of a single application is probably not worth it.

To make a long story short, if the authors managed to adjust the conditions (perhaps by applying a lot slower flow rate) in a way that a nice separation of individual polysomes (equivalent to or even better than SDGC) and of 40S species from the "top" can be achieved, it would win it for me even if it took 1 or 2 h instead of the advertised 15 min.

Reviewer #3:

The manuscript by Yoshikawa and colleagues has significantly improved with the new controls and side-by-side comparison of SEC and SDG. Although the method does not have sufficient resolution to separate n-mer polysomes, it can be useful to quickly separate bulk polysomes from sub-polysomal fractions in a buffer suitable for downstream analysis. Some questions remain, though, that I mention here in order of appearance in the manuscript:

Figure 1: It is interesting to notice that fraction 12 of SDG, containing the 80S-latter half, and fraction 14 of SDG, presumably containing di-somes, show the same profile in SEC (i.e. they seem to be an equilibrated mix of 60S and 80S). Please, discuss how do the authors interpret this. How can the same mix migrate so differently in SDG?

Figure 2 and Results paragraph six: Please, add a reference to support the statement that EDTA treatment increases the effective size of ribosomal subunits. On the same lines, does high salt treatment reduce the effective size of ribosomal subunits? It is curious that both EDTA and high salt result in lighter subunits by SDG, yet the behaviour of subunits on SEC is opposite (compare Figure 2A with Figure 3—figure supplement 1).

Regarding Figure 2, please show a more exposed gel for RPS10 under EDTA treatment. The protein seems absent from any fraction, and one concludes that it is actually degraded during fractionation.

Figure 3: SEC fraction 2 contains polysomes with 4 or more ribosomes, as judged by SDG. This heterogeneity is not observed by EM (EM: b images and Results paragraph nine) where the authors find mainly polysomes of 4 ribosomes. Why?

Please, show a Table of all proteins identified in Figure 6 by cluster.

Figure 7: The fractionation profile of several of the factors is unusual: i) No PABPC1 is found in free RNPs ("smaller complexes"); ii) No EEF2 is present in polysomes; iii) No EJC is found in free RNPs. For all these factors, judging from their functions and their fractionation in SDG, one would expect a different profile. Please discuss, or better provide a parallel analysis of this tissue in SDG.

---

## [Author Response]

[Editors’ note: the author responses to the first round of peer review follow.]

Summary of critique: All three reviewers agreed RM-SEC could be a valuable addition to translation research, at least for its rapidity and amenability to high-throughput applications, and for the absence of high concentrations of sucrose for downstream cryo-EM applications.

We thank the reviewers for these positive remarks, which highlight the potential value of our new method. We would like to stress that in addition to its potential value for studying translation per se, we believe that it also will be very useful for a wider range of applications studying gene regulation and physiological response mechanisms in cells and tissues that alter the sets of proteins being actively translated. To strengthen the new RM-SEC method, we have now added a considerable amount of additional new data to the revised manuscript, including data showing both EM analysis and MS-based proteomic analysis on the fractionated polysomes.

However, all three also voiced serious concerns with the quality of the data and interpretations of certain results. It was agreed that the inability to resolve different n-mer polysomes from one another is an important drawback of the technique that the authors should attempt to resolve.

We have addressed this important issue of polysome resolution directly in the revised manuscript, providing additional experimental verification of our method. In essence, we demonstrate clearly that the SEC approach is a flexible method that can either be used to separate different size classes of polysomes quickly with higher resolution, or to provide even faster isolation of polysomes (i.e. ~ 10 mins) with lower resolution separation of different size classes of polysomes, depending upon the needs of individual experiments. We show that the degree of resolution obtained is a function of SEC flow rate and elution time. We also show clearly in the revised manuscript that the polysomes isolated by SEC are stable, as well as functional, and that they can thus readily be purified with high resolution using 2 sequential rounds of SEC, which can still be achieved within a total analysis time of 30-40 mins, which is substantially faster than a single round of SDGC analysis. Therefore, researchers addressing different experimental questions can benefit from the increased speed and reproducibility of using uHPLC-based fractionation combined with SEC. We have also included new data from EM and proteomic characterisation of the isolated polysomes, demonstrating the advantages of rapid isolation of polysomes in buffers suitable for immediate downstream analysis (new Figure 3). Finally, we have revised the Discussion section to present a more detailed, objective review of the ability of the respective methods to distinguish different size classes of polysomes and how this relates to different experimental applications. We have tried to present a balanced view highlighting objectively the pros and cons of the SEC as well as the SDGC method.

The very minimal 80S peak is also troubling and it was felt that further work is needed to illuminate the dramatic difference between RM-SEC and SDGC in terms of 80S abundance, as this will be crucial to the acceptance of RM-SEC by the translation field. In this regard, it was felt that you should attempt to demonstrate that RM-SEC will detect an extensive conversion of polysomes to inititation-arrested 80S monosomes engendered by Harringtonin, and to compare the high-salt sensitivity of the 80S peaks in SDGC vs RM-SEC.

We have addressed this in the revised manuscript and compared the preparation of HeLa cell lysates using either Triton X-100, or CHAPS as a solubilising detergent (please see our explanation on using CHAPS from line 96). We found no changes in terms of overall polysome profiles examined by SDGC, which shows a similar high fraction of ribosomes in polysomes, as opposed to 80S monosomes (Figure 1—figure supplement 1). We therefore conclude that the relatively low abundance of the 80S peak, compared with the polysome peak, is not specific to the SEC analysis method but rather is characteristic of the translation profile for the HeLa cells and culture conditions used. It may be that under conditions of rapid cell growth there is a high polysome to monosome ratio (Figure 1—figure supplement 1), although we note that this is also seen in mouse liver tissue and may therefore be a more general feature of many mammalian cells.

Moreover, as shown in Figure 5 in the revised manuscript (Figure 3 in the original manuscript), we clearly detect higher ratios of 80S monosomes to polysomes when the same HeLa cells are grown under conditions of amino acid starvation. We therefore are confident that the data support the view that mammalian cells will typically have a high polysome ratio and that this is not caused by a detection problem associated with the SEC method.

We have also analyzed the ‘high-salt’ sensitivity of the 80S peak in SDGC and RM-SEC as requested by reviewers. It is known that the separation in SEC is sensitive to changes in mobile phase components, such as salts, and therefore, as shown in revised Figure 3—figure supplement 1, the resolution of each peak changes with extracts in high salt concentration. We therefore would view the RM-SEC method as not well suited for addressing the specific issue of the ‘vacant 80S couple’. It can however still be used to isolate polysomes from extracts prepared in high salt concentration. We have discussed these specific issues clearly and highlighted the pros and cons of the respective methods in the revised Discussion.

It is also important to present evidence supporting your assertion that a substantial fraction of polysomes dissociate into 80S monosomes during SDGC, as this calls into question a large body of work in the field based on SDGC, starting with the presentation of your own data from an experiment in which polysomes isolated by RM-SEC were resolved by SDGC.

We examined the possibility raised in the discussion in our original manuscript carefully and found no dissociation of polysomes into 80S monosomes (Figure 3B). We have added the data showing this and have therefore revised the text and discussion accordingly.

In addition, the SDGC separation in Figure 1—figure supplement 2, used to assign ribosomal species in RM-SEC is of poor quality, which seemed likely to have overemphasized contamination of the 80S peak with free 60S subunits using SDGC, and would have to be improved.

To address this, we have re-analyzed the polysome profile from HeLa cell extracts and present these new data in a revised Figure 1. This allows us to assign more accurately the profiles for the elution of the different ribosomal particles fractionated by RM-SEC. We collected 22 SDGC fractions in total, i.e. almost twice as many fractions as analysed in the original manuscript, thereby providing higher resolution. To focus on the 80S species, we took 2 fractions containing 80S (Fractions #11 and 12) and showed that even in the latter half of the 80S peak (Fraction #12), it contained a portion of free 60S subunits. Our findings here therefore are consistent with the previously described resolution issues encountered by other researchers using SDGC, for example see the paper describing Ribosome Profiling (Ingolia NT et al., 2009).

Another drawback is working with cells in which the polysome content is very low, and it was felt that you should examine rapidly growing yeast extracts to remedy this problem-especially considering the importance of yeast as a model system for elucidating translation mechanisms.

As mentioned above, we re-analyzed HeLa cell extracts by SDGC (new Figure 1—figure supplement 1), using the same cell growth conditions that we used throughout the experiments in this manuscript. These data from SDGC analyses clearly show that there is a high polysome content in the cells we have studied. We have focused our studies here on developing methods for the analysis of mammalian polysomes/ribosomes, which is our main area of expertise and research interest. We have therefore revised the title and the text with emphasis on the separation of mammalian polysomes/ribosomes by RM-SEC.

We fully acknowledge the potential importance of also using yeast as a model system for studies on translation regulation, as raised by the reviewers. We have therefore initiated studies on yeast extracts also, but feel strongly that this is best reported in a separate future study. Our pilot studies show that considerable additional work is required to optimise the specific combination of column pore size, flow conditions, salt/detergent etc. in the extracts to provide similar resolution to that we have already optimised here for the mammalian system. As we are not yeast biologists, we would welcome the opportunity to pursue this in future in collaboration with colleagues studying relevant research questions in the yeast system who can benefit from using the SEC approach. Similarly, we are interested to evaluate the application of RM-SEC to analyse also nematode and Drosophila extracts. Meanwhile, we feel that the present manuscript will already be of direct value to the many groups interested in studying polysomes and gene regulatory responses in human and/or mammalian model organisms.

There are also numerous issues with inexplicable results from the analysis of rRNA and ribosomal proteins across the RM-SEC fractions, including: (i) in Figure 1b-EDTA-treated lysate, fraction 8 (40S peak) contains a considerable amount of 18S rRNA but no RPS10.

To address this, we have performed further experiments, adding additional Western blotting data to the revised manuscript using the anti-RPS10 antibody to probe all of the SEC fractions. This shows that RPS10 is dissociated from the 40S subunit to smaller protein complexes after EDTA treatment (Figure 2—figure supplement 2). These data from SEC analysis are consistent with similar observations we obtained by parallel SDGC analysis (revised Figure 2—figure supplement 1). We have now included these data in the revised manuscript.

ii) in Figure 1—figure supplement 5-EDTA-treated lysate, fraction 3 (40S) contains no 28S rRNA but a lot of RPLP0. (iii) in Figure 1—figure supplement 5, there is no peak of Rps10 or 18S rRNA in the 80S peak in fraction 6.

To address this, we have re-analyzed the polysome profiles by Western and northern blotting, using HeLa cell lysates +/- EDTA treatment. This shows by SDGC analysis that one of the RPL proteins, RPL14, is mainly detected in the dissociated 60S particle fractions after EDTA treatment (revised Figure 2—figure supplement 1), and also shows that RPS10 and 18S rRNA are both detected in the 80S peak, (revised Figure 2—figure supplement 1). However, as discussed above, under the cell growth conditions used, the total amount of 80S is relatively low compared with the polysome fractions.

And in Figure 3C, the redistribution of ACTB mRNA from polysomes to monosomes appears to be nearly as dramatic as that seen for uL23 mRNA, necessitating quantification of these mRNA distributions from replicates.

We have now quantitated the mRNA signals and added these data in the revised Figure 5.

The discrepancy between the results in Figure 4 and those published by Gao et al., 2015 could be explored further by comparing in your their own hands the results obtained using SDGC and RM-SEC; although this would not be considered essential for acceptance.

We have deleted the data shown as Figure 4 in the original manuscript and replaced this figure with the new data we have obtained, derived from analysis of the fed mouse liver only. We have also now added data showing direct, MS-based proteomic analysis of fractions separated by RM-SEC using the fed mouse liver. These new data are presented and described in revised Figure 6 and 7.

Finally, it is necessary to substantiate your claim that the results of SDGC are less reproducible than from RM-SEC.

To address this, we have performed a detailed analysis of replicate SDGC and SEC profiles and calculated Pearson correlation coefficients to compare the reproducibility from these respective RM-SEC and SDGC analyses. These data are now included in revised Figure 4. We have also revised the text to provide a balanced discussion of the factors that can potentially affect the reproducibility of SDGC profiles.

Reviewer #1:This study describes the use of HPLC size exclusion chromatography to resolve polysomes from monosomes and free subunits. It appears to have several advantages over sucrose density gradient centrifugation (SDGC), such as greater rapidity and reproducibility, and fractionation in physiological buffers versus sucrose. They present evidence that the technique gives reproducible separations consistent with SDGC, with the exception that the 80S (monosome) peak is dramatically reduced in this technique. They favor the idea that the 80S peak is overemphasized in SDGC at least partly due to artefactual dissociation of polysomes into 80S monosomes during centrifugation.This appears to be a very significant technological advance for the translation field.

We thank the reviewer for these positive comments on RM-SEC. We believe that our RM-SEC will be useful as a technical alternative to the conventional SDGC in terms of rapidity, efficiency, reproducibility and biological aspects.

However, questions surrounding the 80S monosome peak could hinder workers from embracing it as an alternative to SDGC, since determining polysome:monosome ratios as a measure of translational activity is frequently a goal of these experiments. As such, the authors should be asked to substantiate their interpretation by presenting the data cited but not shown for the experiment they describe in the Results and Discussion in which the polysome fraction isolated from HPLC was resolved by SDGC. According to their explanation, the HPLC-polysome fraction should yield a substantial proportion of 80S monosomes when separated in SDGC.

We have calculated the polysome:monosome ratio in revised Figure 5 to show that RM-SEC can be used to provide a measure of translational activity. Moreover, this approach can also be used for polysome profiling to estimate translational efficiency, as a ‘heavy polysome’ fraction is successfully isolated and enriched by RM-SEC (revised Figure 3A, B). We haven’t detected dissociated ribosomal subunits from polysome fractions when analyzed by SDGC; however, this analysis also shows that the polysomes separated by RM-SEC are stable and intact (revised Figure 3B).

In view of recent findings from Heyer and Moore that a large fraction of the monosomes are involved in translation elongation, it seems possible that the vacant 80S couples unbound to mRNA thought to constitute a significant fraction of the 80S peak in SDGC dissociate into free subunits in the HPLC separation. If so, essentially all of the small 80S peak resolved by HPLC would be comprised of translating 80S ribosomes and hence, resistant to high salt; whereas the a substantial proportion of the 80S peak in SDGC is known to dissociate in high-salt owing to the presence of vacant 80S couples. It seems incumbent on the authors to examine this question by comparing the amount of monosomes present in HPLC versus SDGC separations in high salt conditions where vacant couples dissociate into free subunits.

Please see our response in ‘Summary of critique’.

Reviewer #2:The manuscript by the Lammond lab on Ribo mega-SEC describes a quick method to separate polysomal complexes, which can be used as an alternative to the more laborious and classical sucrose gradient method. The SEC method is robust and reproducible, and has the potential to become a benchmark for any lab working on translational control. There are some details, however, that need to be clarified.

We thank the reviewer for their positive comments on the RM-SEC method. We believe that our RM-SEC will be very useful as an alternative to the conventional SDGC approach for specific types of experiments that can benefit from its rapidity, efficiency and high reproducibility.

1) It is striking that the 80S peak is almost invisible in most SEC profiles compared to sucrose gradients. Further, in Figure 1—figure supplement 2 each sucrose gradient peak/fraction from U2OS cells is resolved by SEC, and considerable cross-contamination of 60S and 80S subunits is found. For example, the 80S peak in sucrose gradients is found to contain more 60S than 80S by SEC. This is important. Researchers working on translational control and ribosome biogenesis have made conclusions based on the identity of peaks in sucrose gradients, an identity that has been supported by the use of translation inhibitors. Are these conclusions now wrong? For this reason, I think it is relevant to ascertain the composition of the peaks by other methods (e.g. throw the peak material onto an electron microscopy grid to visualize 60S versus 80S ribosomes or mixed assemblies).

We have now performed additional experiments to re-evaluate carefully how to assign the respective ribosomal particles fractionated by RM-SEC. To do this we have injected onto the SEC column different fractions isolated from an SDGC gradient. We believe these data are conclusive and show also the degree of cross-contamination of 60S in 80S particles (please see our response in ‘Summary of critique’).

2) Regarding the issue above, how would profiles in the presence of Harringtonine (which should increase the 80S peak) look like in sucrose gradients compared with SEC? In these conditions, how does the 80S peak from the sucrose gradient appear in SEC? The experiment could be performed using Hela cells for a direct comparison with Figure 1A.

We believe that the conversion of polysomes to 80S monosomes is well illustrated already by the experiments we have included comparing extracts isolated from cells grown under conditions of amino acid starvation in Figure 5 in the revised manuscript (Figure 3 in the original manuscript).

3) The rRNA profiles do not always coincide with the RP profiles. For example, in Figure 1B-EDTA-treated lysate, fraction 8 (40S peak) contains a considerable amount of 18S rRNA but no RPS10. Similarly, in Figure 1—figure supplement 5-EDTA-treated lysate, fraction 3 (40S) contains no 28S rRNA but a lot of RPLP0.

Please see our response in ‘Summary of critique’.

Reviewer #3:In the presented paper, Yoshikawa et al. present Ribo Mega-SEC, a newly developed approach for the separation and biochemical analysis of polysomes and ribosomal subunits using size Exclusion Chromatography and uHPLC. Based on the presented data I agree that it is an interesting alternative technique to the classical SDG, however, I disagree that this method represents a significant improvement of the polysomal analysis in general, mainly with respect to the efficiency, reproducibility and downstream applications.

We thank the reviewer’s comment saying RM-SEC is an interesting alternative technique to the classical SDGC. We also welcome any constructive, objective criticisms on the RM-SEC method that might help us to improve our method/manuscript.

To clarify, we had not intended to propose that SDGC should be abandoned, or to suggest that it will not continue to be useful in many areas, or that our new SEC-based approach is the best solution in every situation. No method is without limitations, and few methods, if any, provide the best solution in every experimental situation. For many years there has been essentially no alternative to SDGC for the analysis of polysomes and we contend that our new RM-SEC method now provides a useful alternative that has distinct advantages for many types of experiments, particularly for gene expression studies in mammalian cells and tissues. Rather than replacing SDGC methods, therefore, we anticipate that the RM-SEC approach will in future encourage more researchers to tackle polysome analyses in their projects, who currently do not use SDGC.

We have provided here strong, objective arguments based upon reproducible data, which highlight specific advantages provided by the RM-SEC method, including specifically; (1) speed of analysis (2) extremely high reproducibility (3) flexibility and convenience and (4) facilitation of downstream biochemical/structural analyses on the isolated complexes. We believe that we have documented these claims clearly and strengthened them with the additional experiments and data added here to our revised manuscript. We have also reviewed the comments and Discussion presented in the revised manuscript in an effort to present as balanced and objective an overview as possible of the relative pros and cons of both the SDGC and RM-SEC approaches. While we acknowledge that there are prominent groups, experienced in using SDGC as a core part of their studies, who have the expertise and equipment to apply this very effectively, nonetheless, the practical limitations inherent in the SDGC workflow we have experienced have also been widely recognised and commented on previously by many researchers.

For example, we will highlight briefly some of the practical issues we and others have experienced when trying to set up and use the SDGC method. Thus, SDGC requires preparation of sucrose density gradients for each analysis, followed by extended ultracentrifugation and then subsequent fraction collection from each gradient by a dedicated fraction collector. Quite apart from access to the specialised equipment for accurately pouring gradients and fractionating them after ultracentrifugation, this is an inherently time-consuming and lengthy set of procedures, each of which introduces a source of technical variation between experiments, as does the preparation of multiple sucrose solutions used to form the gradient. These steps typically involve 5-6 hours (or longer), compared to as little as 15 min for a single shot RM-SEC analysis, using a single mobile phase buffer.

In reviewing the literature from different groups using SDGC to analyse polysomes, we have noticed that many papers show significant variation in the detailed profiles presented. This is particularly noticeable in the polysome region, if you focus on the x-axis positions of each polysome peak. It is likely that a number of parameters inherent to the SDGC workflow can contribute to this variation, including differences in gradient formation between centrifugation tubes, differences in starting position and collection of fractions etc. In contrast, the advantage of automated fractionation in uHPLC systems is that each peak is eluted *at exactly the same time*, across many sequential experiments, which is a major reason for the demonstrably higher reproducibility of SEC separation.

Finally, we would again highlight that it is an unavoidable fact that the polysomes fractionated by SDGC are isolated in high sucrose solution. Depending on the experiments being undertaken, this sucrose must then be removed by either dialysis, or other procedures, before the polysomes can be used for many types of subsequent biochemical assays or structural analyses (e.g. Electron Microscopy). The requirement for removal of sucrose thus introduces another source of potential technical variation and another time-consuming step in the workflow. As we show here, in contrast, the RM-SEC method can be performed in a buffer without sucrose that is compatible with immediate assay/analysis (including EM analysis) of the isolated complexes. In summary, therefore, we believe it is simply an objective description of the respective workflows to conclude that RM-SEC offers advantages of speed, reproducibility and ease of carrying out downstream analyses, in comparison with SDGC.

Major comments:– Resolution of polysomes by Mega-SEC is very poor – they come in one broad peak, in principle also including the monosome, when compared to beautifully separated n-somes by the conventional SDG.

Please see our response in ‘Summary of critique’.

– Figure 1—figure supplement 2B; the second (80S) and third (60S) peaks are not resolved well enough; the 80S peak contains a very large amount of 60S subunits and vice versa.

Please see our response in ‘Summary of critique’.

– I also disagree with the optimal flow rate and urge the authors to test also a lot slower rates (even 0.5 ml/min is not satisfactory at all); extending the analysis from 15 min to 30 min would still preserve the time-spent advantage but at the same it could dramatically improve the resolution that is a real problem with this technique.

We show the profile when separation is performed using a flow rate of 0.2ml/min, which demonstrates that di-somes can indeed be separated. We have revised the text in the revised manuscript to describe more clearly the effect of flow rate on the resolution of complexes. Thank you for pointing this out.

– According to my own (rather extensive) experience, the amount of polysomes obtained by SDG by these authors is exceptionally low (see Figure 1—figure supplement 2A and 5). It almost looks like a very efficient polysomal run-off, why is that? In Figure 1—figure supplement 5 there is also a surprisingly small peak of 18S and RPS10 in the 80S-containing fractions.

This is a very useful comment and we are grateful to the reviewer for highlighting this point. We have performed more experiments to address this issue, which appears to reflect cell type and growth conditions, rather than separation method. Thus, we have re-analysed the polysome profile by SDGC, using HeLa cells (new Figure 1—figure supplement 1 etc.), which clearly shows higher levels of polysomes, as also seen in the SEC analyses from the same extracts and in the mouse liver tissue. We have therefore used these new data in the revised manuscript instead of the previous data obtained with U2OS cell extracts used in Figure 1—figure supplement 2A in the original manuscript.

– I also disagree that the conventional SDG is troublesome with respect to reproducibility and its usage for downstream applications.

Please see our response above and in ‘Summary of critique’. We agree that with experience and practice good results can be obtained also with SDGC and indeed would also see this being a method that may be particularly useful for some specific types of experiments. Nonetheless, we maintain for the reasons explained that RM-SEC can provide a useful alternative approach that also has specific advantages for certain types of experiments. We hope this is now described in a more balanced fashion in the revised manuscript.

– Figure 3C; I see even more dramatic difference in the ACTB distribution here – both fat peaks are gone?

Please see our response above and in ‘Summary of critique’.

– “[…] note that an alternative study, using SDG, reported that polysomes are disassembled in fasted mouse liver (Gao et al., 2015)." I would expect this to be true but one never knows. Nonetheless, this discrepancy may be an important one, worth pursuing further (examining this phenomenon by both techniques in parallel), as its resolution may bring a real advantage of Mega-SEC over SDG into life.

As our major focus here is to document the RM-SEC method, we have deleted the result comparing the polysomes from fed and fasted mouse liver in this study. Instead we provide new data describing the MS-based proteomic analysis of fed mouse liver to show the application of RM-SEC. We will investigate the issue of how polysomes are affected by starvation in mouse tissues more comprehensively in a separate future study.

In summary, I am not sure what advantage (apart for rough high-throughput screening) this technique may bring to the people planning to analyse polysomes over the SDG technique. For example, the authors should, after the very problematic "resolution issue" has been resolved, analyze the distribution of as many translation factors (eIFs, eEFs and eRFs) as possible across the entire profile under various conditions and show that their distribution changes and that these changes are highly reproducible. This would be a real improvement that would most probably change for example my mind to give the Mega-SEC a try.

We have now significantly extended our characterisation of the RM-SEC method to optimise it for efficient downstream MS-based proteomics analyses on the fractionated complexes and included new data describing such experiments in the revised manuscript. Thus, we present proteomic analysis of all of the fractions separated by RM-SEC using mouse liver tissue across the biological replicates (revised Figure 6 and 7). We have also performed clustering analysis to analyse the fractionated complexes detected by MS and highlighted the identified translation factors, as per the request from the reviewer (revised Figure 6 and 7). We thank the referee for prompting us to include this valuable addition to the manuscript, which helps to illustrate the application of the RM-SEC approach for studies analysing gene expression as well as translation complexes.

[Editors' note: the author responses to the re-review follow.]

The reviewers appreciated the large amount of effort that was expended to address the concerns that were raised with the previous version of the paper, and agree that the paper was considerably strengthened by these efforts. There are however some remaining issues. First, although the results shown in Figures 1A and C provide strong evidence that the SEC technique is capable of resolving polysomes of different sizes, none of the separations presented in the paper actually show a resolution of the single polysome peak into discrete n-mer peaks in the manner that is routinely achieved by SDGC. Identification of the distribution of the different n-mer polysomes by SDGC provides valuable information about the rate of translation initiation and also is crucial in experiments that seek to identify the number of ribosomes per mRNA on particular mRNAs by determining the distribution of individual mRNAs across the different n-mer species, either by RT-PCR or Northern analysis of particular mRNAs or by microarray or RNA-seq analysis across the fractions. It is felt therefore that if the authors can provide in the results an SEC separation in which different n-mer species can be resolved, even if this requires a significantly longer separation time than achieved with the 0.8ml/min flow rate chosen for most experiments, this would greatly increase the likelihood that this new approach would be widely embraced by the translation field and that this paper would be highly cited in the future.

Summary of critique response:

We share the aspiration that our paper will have a major impact and be highly cited in the future. Indeed, we anticipate that our new methodology will have impact well beyond studies specifically on protein translation mechanisms, because it provides an efficient, reproducible and accessible approach for isolating both polysome complexes and other very large subcellular machineries. This should be of value to researchers working across a wide range of cell and molecular biology fields.

We recognise from the reviewers’ comments that there may be specific applications, in particular for detailed mechanistic studies on the protein translation machinery, where it would be useful to distinguish monomer, dimer and higher order polysome complexes. We welcome the opportunity to address this issue directly (raised here and in particular by reviewer #2), and have therefore now included additional new data in our revised manuscript (new Figure 1—figure supplement 3). Our new data demonstrate that we can indeed increase the resolution of separation of polysomes, as well as ribosomal subunits and smaller protein complexes, by fractionating cell extracts on two SEC columns connected in series, *performed in a single integrated experiment*. We have illustrated this in experiments comparing the use of either two 2,000 Å SEC columns, or a 2,000 Å SEC column combined with a 1,000 Å SEC column, run with a flow rate of 0.2 ml/min. The results are shown in the new Figure 1—figure supplement 3, which shows separation of di-some, tri-some and higher order n-mer species, in a single experiment. With regard to the separation of the 40S species from smaller protein complexes (e.g. which may be relevant for experiments involving the analysis of protein translation initiation complexes), we show that fractionating lysates using a 1,000 Å SEC column provides a high resolution separation between 40S and smaller protein complexes (Figure 1—figure supplement 2). Further increases in the resolution of different translation complexes and polysome species may also be possible if the set-up and flow-rates are adjusted further for specialised applications, but we have not pursued this exhaustively here as we concentrate in this manuscript on illustrating the general application of the methodology.

In summary, our new data show that by employing an experimental setup using two SEC columns connected in series in a single shot analysis already provides a nice separation across a very wide range of size classes, ranging from larger polysomes down to smaller protein complexes (Figure 1—figure supplement 3). In the revised manuscript, as well as showing this new figure, we have added a short description of the new data in the Results section and also added further comments to the Discussion section regarding the flexibility this provides to tune the separation of cell and tissue extracts to meet the specific requirements of different experimental applications.

Reviewer #1:I am satisfied that the addition of new data revisions of text and figures have adequately addressed all of the major concerns with the previous version. There are however some errors in citing figures or the relevant panels of figures that need to be corrected as follows:– Results section, paragraph three: cite specific panels of Figure 1 and Figure 1—figure supplement 2 rather than the entire figures to refer to specific results.– Results section, paragraph ten.: Again, cite panels A and B of Figure 3—figure supplement 1 where appropriate rather than just citing the whole figure. Check entire manuscript for proper citations of specific panels of figures throughout.– Results section, paragraph eighteen: shouldn't this cite Figure 6—figure supplement 3A vs 3B?

We thank reviewer #1 for their positive comments and for pointing out these minor errors. We have now corrected them as suggested.

Reviewer #2:The manuscript has significantly improved. The authors carried out a lot of new valuable experiments that have convinced me this technique can be used as an alternative to the classical SDGC in some – but not all – applications. Nonetheless, I still have many concerns – outlined below – that should be addressed before this paper introducing a novel technique can be accepted for publication.[…] To make a long story short, if the authors managed to adjust the conditions (perhaps by applying a lot slower flow rate) in a way that a nice separation of individual polysomes (equivalent to or even better than SDGC) and of 40S species from the "top" can be achieved, it would win it for me even if it took 1 or 2 h instead of the advertised 15 min.

We thank the reviewer for their comments and also for providing their perspective on using the Ribo Mega-SEC method as an alternative approach to SDG for studying translation initiation and termination. While we recognise this is helpful, we would hope that reviewer #2 can nonetheless appreciate that our method can also be of value to many other researchers, beyond the mechanisms of translation field, whose requirements and experimental systems are different. For example, while reviewer #2 highlights that in their view/situation, "*Speed is not all what matters*", we are confident that in many other laboratories, factors such as speed, reproducibility and efficiency will matter. Indeed, this was what led us to develop the Ribo Mega-SEC method. As we describe in the revised manuscript, the Ribo Mega-SEC method is flexible and can be adapted further to optimise its application in a wide range of experimental scenarios.

Regarding the P/M ratio, we already show analysis of the P/M ratio in Figure 5 of the previous version of the manuscript, where we monitor the changes in translation before/after amino acid starvation. To expand on this point and illustrate that the Ribo Mega-SEC method can be used to calculate the P/M ratio, we now include an additional new figure in the revised manuscript, showing a profile and P/M ratio using the 0.2 ml/min flow rate (new Figure 1D). The P/M ratio can be calculated readily from the profile obtained using either this slower flow rate of 0.2 ml/min, or using the faster 0.8 ml/min flow rate we find to be suitable for most applications.

In addition, we have included in the revised manuscript a new figure (Figure 1—figure supplement 3), as described above in our response to the general comments from the editors, which also addresses this issue. In this new figure we show a higher resolution profile derived from performing separations on two SEC columns connected in series, in a single experiment, using a flow rate of 0.2 ml/min.

Reviewer #3:The manuscript by Yoshikawa and colleagues has significantly improved with the new controls and side-by-side comparison of SEC and SDG. Although the method does not have sufficient resolution to separate n-mer polysomes, it can be useful to quickly separate bulk polysomes from sub-polysomal fractions in a buffer suitable for downstream analysis.

We thank the reviewer for their positive remarks and agree that our Ribo Mega-SEC method can be used broadly in the research community.

Some questions remain, though, that I mention here in order of appearance in the manuscript:Figure 1: It is interesting to notice that fraction 12 of SDG, containing the 80S-latter half, and fraction 14 of SDG, presumably containing di-somes, show the same profile in SEC (i.e. they seem to be an equilibrated mix of 60S and 80S). Please, discuss how do the authors interpret this. How can the same mix migrate so differently in SDG?

We have not pursued this point experimentally. Our opinion is that the observation most likely reflects the high abundance of the 60S ribosomal subunit. We detect a small portion of the abundant 60S subunit as a cross-contamination across all of the fractions sedimenting faster than 60S. There is also a small cross-contamination of 80S detected together with di-somes in fraction 14. Note that the peak heights of 80S and 60S obtained by injecting fraction 14 onto SEC (Figure 1C) is markedly different to those seen by injecting fraction 12 onto SEC (Figure 1B) and these differences in intensity are consistent between SEC and SDG analyses (Figure 1A).

Figure 2 and Results paragraph six: Please, add a reference to support the statement that EDTA treatment increases the effective size of ribosomal subunits. On the same lines, does high salt treatment reduce the effective size of ribosomal subunits? It is curious that both EDTA and high salt result in lighter subunits by SDG, yet the behaviour of subunits on SEC is opposite (compare Figure 2A with Figure 3—figure supplement 1).

We have added a reference to support this statement as suggested. We have also included discussion of the effect of high salt treatment on separation by SDG and SEC.

Regarding Figure 2, please show a more exposed gel for RPS10 under EDTA treatment. The protein seems absent from any fraction, and one concludes that it is actually degraded during fractionation.

We have now replaced the image with a longer exposure, as suggested. In addition, we have added an explanation for the number of fractions collected in total (48 fractions, new Figure 2—figure supplement 2). Some fractions were used for Western and northern blotting.

Figure 3: SEC fraction 2 contains polysomes with 4 or more ribosomes, as judged by SDG. This heterogeneity is not observed by EM (EM: b images and Results paragraph nine) where the authors find mainly polysomes of 4 ribosomes. Why?

We thank referee #3 for pointing this out. We have checked the images taken from SEC fraction 2 and accordingly have replaced the image shown with a more representative example and also revised the text to describe the images more accurately.

Please, show a Table of all proteins identified in Figure 6 by cluster.

We have now added a supplementary table listing all of the proteins identified in Figure 6 classified by cluster.

Figure 7: The fractionation profile of several of the factors is unusual: i) No PABPC1 is found in free RNPs ("smaller complexes"); ii) No EEF2 is present in polysomes; iii) No EJC is found in free RNPs. For all these factors, judging from their functions and their fractionation in SDG, one would expect a different profile. Please discuss, or better provide a parallel analysis of this tissue in SDG.

We have now added references to support these results and have included discussion of the data.